# DeepEyes: Incentivizing "Thinking with Images" via Reinforcement Learning

**Ziwei Zheng**[1,2*], **Michael Yang**[1*], **Jack Hong**[1*], **Chenxiao Zhao**[1*†],
**Guohai Xu**[1‡], **Le Yang**[2‡], **Chao Shen**[2], **Xing Yu**[1]

[1]Xiaohongshu Inc., [2] Xi'an Jiaotong University
[*] Equal contribution, Random order [†] Main Code Contributor [‡] Corresponding Author

Project Homepage

{chenxiao2, xuguohai}@xiaohongshu.com, yangle15@xjtu.edu.cn,
ziwei.zheng@stu.xjtu.edu.cn, {yangminghao199,jaaackhong}@gmail.com

## Abstract

Large Vision-Language Models excel at multimodal understanding but struggle to deeply integrate visual information into their predominantly text-based reasoning processes, a key challenge in mirroring human cognition. To address this, we introduce *DeepEyes*, a model that learns to "think with images", trained end-to-end with reinforcement learning without requiring pre-collected reasoning data for cold-start supervised fine-tuning (SFT). Notably, this ability emerges natively, leveraging the model's own grounding capability as an intrinsic function rather than relying on external specialized models or APIs. We enable this capability through active perception, where the model learns to strategically ground its reasoning in visual information, guided by a tailored data selection and reward strategy. *DeepEyes* achieves significant performance gains on general perception and reasoning benchmarks and also demonstrates improvement in grounding, hallucination, and mathematical reasoning tasks. Interestingly, we observe the distinct evolution of active perception from initial exploration to efficient and accurate exploitation, and diverse thinking patterns that closely mirror human visual reasoning processes. Code is available at https://github.com/Visual-Agent/DeepEyes.

## 1 Introduction

Recent advances in Vision-Language Models (VLMs) have enabled deeper reasoning over multimodal inputs by adopting long Chain-of-Thought (CoT) approaches (Team et al., 2025a;b; Guo et al., 2025b), allowing these models to handle more complex tasks. However, these models still primarily rely on text-based reasoning, with their thought processes largely confined to the language modality. In contrast, human reasoning naturally combines vision and cognition, thinking with images by extracting information through sequential visual fixations, which support more accurate perceptual decision-making, which was essential for survival in early human evolution (Najemnik & Geisler, 2005). While some recent works have proposed pre-defined workflow-based strategies to incorporate visual information into CoT reasoning (Shao et al., 2024a; Sun et al., 2024), the modular designs suffer from suboptimal performance (Ross et al., 2011).

In a recent milestone, the OpenAI o3 model (OpenAI, 2025) has successfully integrated visual information as a dynamic element in the reasoning process. The o3 transcends the language-modality confinement by extending reasoning capability to "thinking with images" like humans. Additionally, it resolves the coordination limitations by combining textual CoT and image manipulation tools in a naturally interleaved fashion during the CoT process. This approach enables a new axis for test-time

---

[*]Work done during Ziwei's internship at Xiaohongshu. The specific contribution of co-first authors is shown in the Appendix.C.

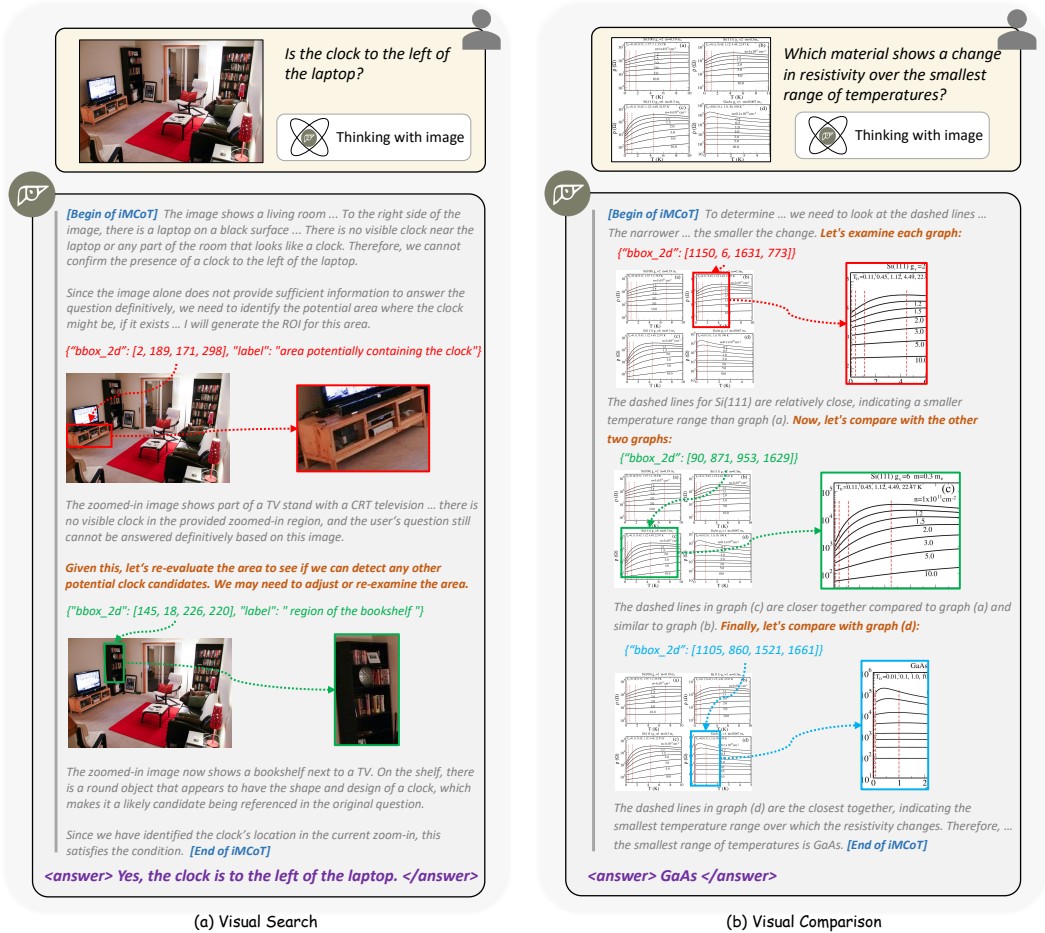

Figure 1: **Interleaved Multi-modal Chain-of-Thought (iMCoT).** *DeepEyes* is incentivized to perform active perception throughout the reasoning process with end-to-end reinforcement learning.

compute scaling by seamlessly integrating visual and textual reasoning, representing a meaningful advancement toward true multimodal reasoning. However, the inner mechanism remains undisclosed to the open-source community.

In this paper, we introduce ***DeepEyes***, a model with "thinking with images" ability, which is incentivized via end-to-end reinforcement learning. This capability emerges natively without relying on separate specialized models and is directly guided by outcome rewards, eliminating the need for cold-start supervised fine-tuning used in previous methods. Specifically, we encapsulate the model's grounding ability in an active perception mechanism, enabling it to gather information from the original image within an agentic framework. As shown in Figure 1, the model adaptively generates image grounding coordinates and crops relevant regions, which are then incorporated into the ongoing reasoning trajectory. This supports an interleaved Multimodal Chain-of-Thought (iMCoT), where visual and textual reasoning are seamlessly integrated.

In early attempts, we observe that the model struggles to effectively utilize its active perception capability. Specifically, it is reluctant to perform image zoom-ins and even when it does, the exploration often selects suboptimal regions. This results in low rewards and unstable training dynamics. To address these issues, we propose a data selection mechanism to choose training samples based on their potential to encourage active perception behavior. Additionally, we design a reward strategy that assigns a conditional bonus to the trajectories that successfully complete their tasks through active perception. Our ablation studies validate that these two strategies are crucial for optimizing the efficiency and accuracy of active perception.

Without supervised fine-tuning (SFT) for intermediate reasoning steps, we observe the model's active perception strategy evolving through three distinct stages during RL training: (1) initial, ineffective

exploration; followed by (2) frequent and effective application of the capability; and finally, (3) a mature, selective, and efficient approach yielding high performance. This progression demonstrates the model's growing mastery of its visual reasoning capabilities through active perception. Additionally, diverse iMCoT reasoning patterns emerge, such as visual search for small or hard-to-recognize objects, visual comparisons across different regions, visual confirmation to eliminate uncertainty, and hallucination mitigation by focusing on details. These diverse reasoning behaviors closely resemble human cognitive processes, thereby enhancing the system's overall multimodal capabilities.

Experimental results show that *DeepEyes* can significantly boost performance on multiple visual perception and reasoning tasks. For high-resolution benchmarks, *DeepEyes* with a 7B model achieves an accuracy of 90.1% (+18.9 %) on $V^*$, and improves HR-Bench-4K and HR-Bench-8K by 6.3% and 7.3%, respectively. In addition, *DeepEyes* also improves multimodal capabilities on a wide range of tasks such as visual grounding, hallucination mitigation, and mathematical problem solving. The main contributions are summarized as follows:

- We incentivize and enhance the ability of thinking with images via end-to-end reinforcement learning, forming iMCoT that seamlessly blends visual-textual reasoning without requiring cold-start SFT or separate specialized models as external tools.

- To better incentivize the model's interleaving reasoning, we introduce an active-perception data selection mechanism and a tailored reward strategy that promote grounding-assisted problem solving. Experiments show that both components significantly advance iMCoT.

- We reveal the intriguing RL training dynamic of iMCoT, where active perception behavior undergoes distinct stages, evolving from initial exploration to efficient and accurate exploitation. We also observe diverse reasoning patterns, such as visual search, comparison, and confirmation.

## 2 RELATED WORK

**Multi-modal Large Language Models.** Multimodal large language models (MLLMs) have evolved from early systems that loosely combined vision encoders with language models into more integrated architectures through joint training. Methods such as BLIP-2 (Li et al., 2023b) and LLaVA (Liu et al., 2023b;a) align visual and linguistic modalities by projecting image features into the latent space of frozen LLMs using query transformers or lightweight projectors, enabling tasks like visual question answering and instruction following. To address resolution constraints, approaches like AnyRes (Liu et al., 2024a; Chen et al., 2024a) allow for flexible image sizes and enhanced visual fidelity. These advances have led to strong open-source models, including the LLaVA (Liu et al., 2024b; Guo et al., 2024; Zhang et al., 2025b; Lin et al., 2023; Li et al., 2023a), Qwen-VL (Bai et al., 2023; Wang et al., 2024b; Yang et al., 2024), and InternVL (Chen et al., 2024c; Gao et al., 2024; Lu et al., 2025) series. Concurrently, large-scale models like Flamingo (Alayrac et al., 2022), mPLUG-Owl (Ye et al., 2023; 2024b;a), and GPT-4V (Yang et al., 2023) aim to unify vision-language understanding, incorporating mechanisms such as mixture-of-experts (Shu et al., 2024; Li et al., 2025c; Shen et al., 2024b) or image generation (Xie et al., 2024; Xu et al., 2025). However, these models lack reasoning capabilities like Chain-of-Thought and test-time scalability (Muennighoff et al., 2025; Zhang et al., 2025a; Chen et al., 2024b), and still decouple perception from reasoning.

**Vision-language Model Reasoning.** Existing Multimodal Chain-of-Thought (MCoT) reasoning methods fall into two main categories. Early approaches rely on predefined workflows or auxiliary models (Liu et al., 2024c; Mondal et al., 2024; Luo et al., 2024), often focusing on region-of-interest localization (Wu & Xie, 2024; Fu et al., 2025; Wei et al., 2025; Li et al., 2025b), latent feature regeneration (He et al., 2024; Bigverdi et al., 2024), and external knowledge integration (Sun et al., 2024; Li et al., 2025a) to improve interoperability. Inspired by the extensive research on the long CoT in LLMs (Guo et al., 2025a), RL-based reasoning approaches have been increasingly explored in MLLMs (Meng et al., 2025; Peng et al., 2025; Shen et al., 2025). These methods predominantly extend text-only reasoning capabilities to a range of multimodal tasks such as spatial reasoning (Zhou et al., 2025), object recognition (Liu et al., 2025b), semantic segmentation (Liu et al., 2025a), and video tasks (Zhao et al., 2025a;b). Unlike methods that hard-code pipelines or simply extend text-only CoT, our approach lets the model autonomously decide when and how to use visual input. Guided by outcome rewards, it adapts visual exploration for a more flexible reasoning process.

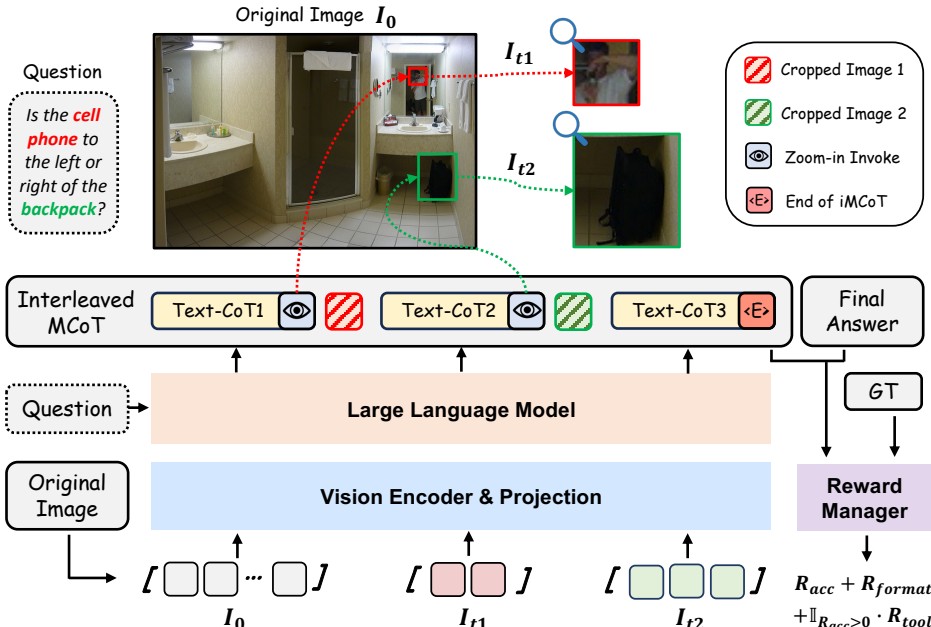

Figure 2: **Overview of *DeepEyes*.** Our model itself decides whether to perform a second perception via zoom-in by generating grounding coordinates and cropping relevant regions, or to answer directly.

## 3 METHOD

### 3.1 DEEPEYES

*DeepEyes* is a unified multimodal large language model that is capable of "thinking with images" through an iMCoT reasoning process. The ability is inherited from the model's native capability of visual grounding and action decision planning, and further incentivized and enhanced via end-to-end RL training using outcome reward signals, eliminating the need for cold-start supervised fine-tuning.

As illustrated in Figure 2, given a user question and an image $I_0$ as input, *DeepEyes* can autonomously decide, after each textual CoT reasoning step, whether to generate an answer directly or perform an image zoom-in for further inspection. The zoom-in operation takes a list of bounding box coordinates as input and outputs the cropped images within the specified regions. The returned crops, such as $I_{t1}$ and $I_{t2}$, are appended to the ongoing trajectory, enabling the model to reason over all previous context. *DeepEyes* can perform active perception as many times as needed before concluding a final answer. This iterative interaction enables fine-grained perception, especially when the relevant object in the image is small, blurry, or difficult to recognize. During the RL training stage, the reward optimization policy gradient is applied to the entire trajectory, allowing all textual CoTs and action decision planning to be jointly optimized in an end-to-end manner.

Compared to previous works based on workflows or pure text reasoning, our iMCoT offers several significant advantages. (1) **Simplicity in Training.** Previous workflow-based methods Wu & Xie (2024); Li et al. (2025b) depend on substantial SFT data, which is challenging to acquire, while our iMCoT only requires question-answer pairs, reducing data collection complexity. (2) **Enhanced Generalizability.** Workflow-based models are constrained by their task-specific manual design, which hinders their generalization to other tasks. In contrast, our iMCoT exhibits robust generalization capabilities as it learns to dynamically select optimal reasoning processes across diverse tasks through reinforcement learning. (3) **Global Optimization.** Our iMCoT enables joint optimization through end-to-end training, which allows the system to be optimized towards a global optimum. In contrast, optimizing each component separately typically leads to sub-optimal performance. (4) **Multimodal Integration.** Compared to pure text-based thinking, our iMCoT naturally interleaves visual and textual information, combining visual elements with textual reasoning to achieve more accurate perceptual decision-making. (5) **Native Tool Calling.** We encapsulate the model's native grounding

capability as an internal tool to enable active perception, allowing implicit optimization that previous external-tool paradigms cannot achieve.

## 3.2 AGENTIC REINFORCEMENT LEARNING

**Rollout Formulation.** In traditional RL with text-only CoT, the Markov Decision Process (MDP) defines the state as the input prompt tokens together with all tokens generated by the model up to the current step. The action is defined as the next token in the sequence. In contrast, agentic RL extends this formulation by introducing observation tokens, which come from external function calls rather than the model itself. These observation tokens are appended to the ongoing rollout sequence and fed back into the model as input for the subsequent step. We formalize the MDP definition for iMCoT as follows. At each step $t$, the state $s_t$ of iMCoT is defined as:

$$s_t = \{(X_0, I_0), (X_1, I_1), \ldots, (X_t, I_t)\} = \{\mathbf{X}_{\leq t}; \mathbf{I}_{\leq t}\}, \tag{1}$$

where $\mathbf{X}_{\leq t} = \{X_1, \ldots, X_t\}$ represents the accumulated sequence of text tokens before step $t$, and $\mathbf{I}_{\leq t} = \{I_1, \ldots, I_t\}$ represents the image observation tokens before step $t$. We omit other related special tokens that are not generated by VLM itself for simplicity. Given the state $s_t$, the action $a_t \sim \pi_\theta(a \mid s_t)$ is sampled from the VLM policy $\pi_\theta$, serving as the next input token. This iMCoT continues to interleave until either an answer is generated or the maximum number of active perceptions is reached. Note that text tokens $\mathbf{X}_{\leq \mathbf{t}}$ and image tokens $\mathbf{I}_{\leq \mathbf{t}}$ are interleaved in the states.

**Reward Design.** In multimodal environments, sparse, outcome-driven rewards are essential for guiding vision-language models toward effective reasoning and decision-making. Because intermediate visual actions lack step-level supervision, we evaluate the entire reasoning trajectory based on the final outcome and the presence of meaningful active perception.

The total reward consists of three parts: an accuracy reward $R_{\text{acc}}$, a format reward $R_{\text{format}}$, and a conditional bonus $R_{\text{tool}}$. Accuracy measures whether the final answer is correct, while formatting penalizes poorly structured outputs. The conditional bonus is granted only when the answer is correct and at least one active perception step is triggered:

$$R(\tau) = R_{\text{acc}}(\tau) + R_{\text{format}}(\tau) + \mathbb{I}_{R_{\text{acc}}(\tau) > 0} R_{\text{tool}}(\tau), \tag{2}$$

where $\mathbb{I}_{R_{\text{acc}}(\tau) > 0}$ equals 1 if the accuracy reward is positive. Conditioning this bonus on a correct answer promotes perception-aware reasoning while discouraging unnecessary actions (see Section 4.3).

**Optimization.** We adopt Group Relative Policy Optimization (GRPO) (Shao et al., 2024b), which has been proven to be effective for diverse tasks. For multi-turn reasoning trajectories, we apply a token-wise loss mask to ignore loss on observation tokens not generated by the model.

## 3.3 TRAINING DATA CURATION

A key challenge in training our model via RL is ensuring initial sampling efficiency without an SFT cold start. To address this, we designed a data curation strategy to construct a corpus that is both diverse and specifically targeted to bootstrap effective active perception behavior from the outset.

**Data Collection.** To construct a robust training corpus, we combine three complementary sources targeting key capabilities: the $V^*$ training set (Wu & Xie, 2024) for fine-grained perception, chart data from ArxivQA (Li et al., 2024b) for task and image diversity, and the ThinkLite-VL (Wang et al., 2025b) dataset to strengthen challenging reasoning. This combination provides a multifaceted foundation for our iMCoT framework, with further details available in Appendix B.

**Data Selection.** We employ a multi-stage filtering pipeline to curate a dataset aimed at strengthening grounding-assisted visual reasoning. The process begins with *difficulty curation*, where we use Qwen2.5-VL-7B (Bai et al., 2025) to assess question difficulty, removing samples that are either too trivial (100% Acc.) or overly challenging (0% Acc.). Next, we standardize all questions into an open-ended format and perform *data verification* to eliminate incorrectly labeled samples. The final stage applies a *perception-utility filter*, retaining only samples solvable via active perception with ground-truth regions, thereby maximizing informational gain and boosting initial RL sampling efficiency without an SFT cold start. This last filter is applied only to the fine-grained perception data; chart and general reasoning data are preserved in their original, rigorously processed form. The resulting dataset is well-suited for training models with strong interleaved reasoning capabilities.

Table 1: **Results on High-Resolution Benchmarks.** E2E indicates whether the model is end-to-end, requiring no manually defined workflow. * denotes reproduced results.

| Model | E2E | Param Size | $V^*$ Bench | | | HR-Bench 4K | | | HR-Bench 8K | | |
|---|---|---|---|---|---|---|---|---|---|---|---|
| | | | Attr | Spatial | Overall | FSP | FCP | Overall | FSP | FCP | Overall |
| GPT-4o Achiam et al. (2023) | ✓ | - | - | - | 66.0 | 70.0 | 48.0 | 59.0 | 62.0 | 49.0 | 55.5 |
| o3 OpenAI (2025) | ✓ | - | - | - | 95.7 | - | - | - | - | - | - |
| SEAL Wu & Xie (2024) | ✗ | 7B | 74.8 | 76.3 | 75.4 | - | - | - | - | - | - |
| DyFo Li et al. (2025b) | ✗ | 7B | 80.0 | 82.9 | 81.2 | - | - | - | - | - | - |
| ZoomEye Shen et al. (2024a) | ✗ | 7B | 93.9 | 85.5 | 90.6 | 84.3 | 55.0 | 69.6 | 88.5 | 50.0 | 69.3 |
| LLaVA-OneVision Li et al. (2024a) | ✓ | 7B | 75.7 | 75.0 | 75.4 | 72.0 | 54.0 | 63.0 | 67.3 | 52.3 | 59.8 |
| Qwen2.5-VL* Bai et al. (2025) | ✓ | 7B | 73.9 | 67.1 | 71.2 | 85.2 | 52.2 | 68.8 | 78.8 | 51.8 | 65.3 |
| Pixel-Reasoner Su et al. (2025) | ✓ | 7B | 83.5 | 76.3 | 80.6 | 86.0 | 60.3 | 72.9 | 80.0 | 54.3 | 66.9 |
| Qwen2.5-VL* Bai et al. (2025) | ✓ | 32B | 87.8 | 88.1 | 87.9 | 89.8 | 58.0 | 73.9 | 84.5 | 56.3 | 70.4 |
| **DeepEyes** | ✓ | 7B | 91.3 | 88.2 | 90.1 | 91.3 | 59.0 | 75.1 | 86.8 | 58.5 | 72.6 |
| $\Delta$ (*vs* Qwen2.5-VL 7B) | - | - | +17.4 | +21.1 | +18.9 | +6.1 | +6.8 | +6.3 | +10.0 | +6.8 | +7.3 |

Table 2: **Results on General Perception and Reasoning Benchmark** MME-RealWorld-Lite.

| Model | Param Size | Overall | Perception | | | | | Reasoning | | | |
|---|---|---|---|---|---|---|---|---|---|---|---|
| | | | OCR | RS | DT | MO | AD | OCR | DT | MO | AD |
| LLaVA-OneVision Li et al. (2024a) | 7B | 43.7 | 80.0 | 40.0 | 56.0 | 31.7 | 39.4 | 65.0 | 33.0 | 38.0 | 32.0 |
| Qwen2.5-VL Bai et al. (2025) | 7B | 42.3 | 87.6 | 32.7 | 83.0 | 27.3 | 30.0 | 72.0 | 62.0 | 28.7 | 23.0 |
| Qwen2.5-VL Bai et al. (2025) | 32B | 45.6 | 87.2 | 40.7 | 83.0 | 29.5 | 40.7 | 74.0 | 60.0 | 27.3 | 29.5 |
| Pixel-Reasoner Su et al. (2025) | 7B | 49.7 | 89.6 | 52.0 | 86.0 | 38.9 | 30.9 | 71.0 | 72.0 | 46.0 | 32.5 |
| **DeepEyes** | 7B | 53.2 | 90.0 | 52.7 | 89.0 | 43.3 | 33.4 | 76.0 | 69.0 | 44.0 | 35.0 |
| $\Delta$ (*vs* Qwen2.5-VL 7B) | - | +10.9 | +2.4 | +20.0 | +6.0 | +16.0 | +3.4 | +4.0 | +7.0 | +15.3 | +12.0 |

Table 3: **Results on Grounding and Hallucination Benchmarks.** * denotes reproduced results.

| Model | Param Size | refCOCO | refCOCO+ | refCOCOg | ReasonSeg | POPE | | | |
|---|---|---|---|---|---|---|---|---|---|
| | | | | | | Adversarial | Popular | Random | Overall |
| LLaVA-OneVision Li et al. (2024a) | 7B | - | - | - | - | - | - | - | 88.4 |
| Qwen2.5-VL Bai et al. (2025) | 7B | 90.0 | 84.2 | 87.2 | - | - | - | - | - |
| Qwen2.5-VL* Bai et al. (2025) | 7B | 89.1 | 82.6 | 86.1 | 68.3 | 85.9 | 86.5 | 87.2 | 85.9 |
| **DeepEyes** | 7B | 89.8 | 83.6 | 86.7 | 68.6 | 84.0 | 87.5 | 91.8 | 87.7 |
| $\Delta$ (*vs* Qwen2.5-VL 7B) | - | +0.7 | +1.0 | +0.6 | +0.3 | -1.9 | +1.0 | +4.6 | +1.8 |

Table 4: **Results on Challenging Reasoning Benchmarks.** * denotes reproduced results, and † denotes results taken from (Zhu et al., 2025).

| Model | Param Size | MathVista | MathVerse | MathVision | WeMath | DynaMath | LogicVista |
|---|---|---|---|---|---|---|---|
| LLaVA-OneVision Li et al. (2024a) | 7B | 58.6† | 19.3† | 18.3† | 20.9† | - | 33.3† |
| Qwen2.5-VL Bai et al. (2025) | 7B | 68.2 | 49.2 | 25.1 | 35.2† | - | 44.1† |
| Qwen2.5-VL* Bai et al. (2025) | 7B | 68.3 | 45.6 | 25.6 | 34.6 | 53.3 | 45.9 |
| **DeepEyes** | 7B | 70.1 | 47.3 | 26.6 | 38.9 | 55.0 | 47.7 |
| $\Delta$ (*vs* Qwen2.5-VL 7B) | - | +1.9 | +1.7 | +1.0 | +4.3 | +1.7 | +1.8 |

# 4 EXPERIMENT

## 4.1 SETUPS

**Baselines and Benchmarks.** To comprehensively assess the effectiveness of *DeepEyes*, we compare it against three categories of baselines: (1) advanced *proprietary* models, including OpenAI GPT-4o (Achiam et al., 2023) and o3 (OpenAI, 2025); (2) state-of-the-art *open-source* models, such as LLaVA-OneVision (Li et al., 2024a) and Qwen2.5-VL (Bai et al., 2025); and (3) approaches explicitly designed with *workflows*, such as SEAL (Wu & Xie, 2024), DyFo (Li et al., 2025b) and ZoomEye (Shen et al., 2024a). Since tasks requiring fine-grained visual understanding naturally highlight the strengths of iMCoT, we first evaluate *DeepEyes* on high-resolution benchmarks. Then, we assess *DeepEyes* on grounding and hallucination benchmarks to show improvements brought by iMCoT on general visual capabilities. We also adopt general reasoning benchmarks to verify its effectiveness.

**Training Details.** We train Qwen2.5-VL-7B with GRPO for 80 iterations on H100 GPUs. Each batch samples 256 prompts, with 16 rollouts per prompt, up to a maximum of 6 times of active perceptions. We set the KL coefficient to 0.0 and define the maximum response length as 20480 tokens.

## 4.2 MAIN RESULTS

**High-Resolution Benchmarks.** High-resolution benchmarks, such as $V^*$ (Wu & Xie, 2024) and HR-Bench (Wang et al., 2025a), contain very large images (2K–8K) with small target objects, making accurate localization challenging for VLMs. As shown in Table 1, our model significantly outperforms existing open-source methods, including complex pipelines (Wu & Xie, 2024; Li et al., 2025b; Shen et al., 2024a), achieving $18.9\%$ and $7.3\%$ gains over Qwen2.5-VL 7B on $V^*$ and HR-Bench 8K, respectively. This demonstrates that simple RL can effectively unlock high-resolution visual reasoning without elaborate pipelines.

**General Perception and Reasoning Benchmark.** As shown in Table 2, our 7B model delivers top performance on MME-RealWorld-Lite (Zhang et al., 2024b). It surpasses both the 7B and even 32B versions of Qwen2.5-VL, demonstrating superior real-world perception and reasoning.

**Grounding and Hallucination Benchmarks.** Furthermore, the multimodal CoT enhances general visual capabilities. Evaluated on grounding (refCOCO/refCOCO+ (Caesar et al., 2018), refCOCOg (Kazemzadeh et al., 2014), ReasonSeg (Lai et al., 2024)) and hallucination (POPE (Li et al., 2023c)) benchmarks, our model achieves higher grounding accuracy and substantially reduces hallucinations (Table 3). This improvement stems from our model's ability to focus on regions of interest during visual reasoning and analyze cropped areas in detail, enabling more confident verification of object presence. These results show that iMCoT not only boosts high-resolution perception but also enhances overall visual reliability with a more thorough verification mechanism.

**Challenging Reasoning Benchmarks.** We further evaluate our model on MathVista (Lu et al., 2023), MathVerse (Zhang et al., 2024a), MathVision (Wang et al., 2024a), WeMath (Qiao et al., 2024), DynaMath (Zou et al., 2024), and LogicVista (Xiao et al., 2024) in Table 4. Benefiting from the integrated chain-of-thought mechanism, our model achieves consistent performance improvements across these challenging multimodal reasoning benchmarks, including mathematical problem-solving.

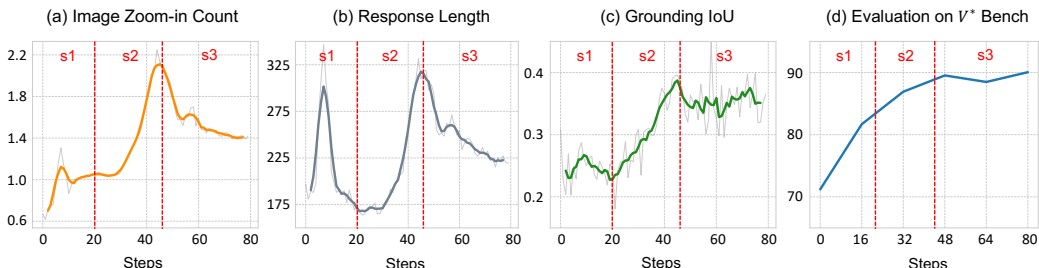

Figure 3: **Training dynamics of *DeepEyes* on $V^*$. s1/2/3 represent different stages.

## 4.3 KEY FINDINGS: FROM CASUAL USER TO PROFICIENT VISUAL REASONER

**Training Dynamics.** To better understand the model's behavior during end-to-end reinforcement learning, we analyze its performance on fine-grained data $V^*$. Since fine-grained data includes ground-truth bounding boxes closely aligned with target answers, we quantify the quality of the model's visual grounding using Intersection-over-Union (IoU). In Figure 3, a clear evolution emerges in how the model leverages active perception. This progression unfolds in three stages, reflecting increasingly effective integration of active perception into reasoning:

• **Stage 1: Initial Exploration (Steps 0–20)** The model starts following system prompts to access additional visual cues, but lacks a coherent strategy. Action count and response length rise, reflecting exploratory behavior, while low grounding IoU shows repeated attempts without successfully linking retrieved information to the visual context. A sharp drop in response length between steps 8 and 20 indicates it is streamlining descriptions while acquiring basic active perception skills.

• **Stage 2: High-Frequency Engagement (Steps 20–45)** The model enters a phase of intensive active perception, repeatedly leveraging visual information to boost accuracy and reward. Key metrics, including grounding IoU, improve, while longer responses and frequent visual interactions suggest a "broad sweep" strategy: the model externalizes reasoning by over-querying the environment. This stage reflects growing recognition of active perception's value, though efficiency remains suboptimal.

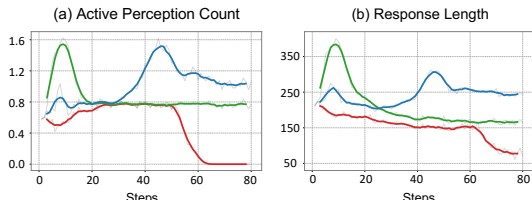

Figure 4: Training dynamics w.r.t. tool reward.

Table 5: Evaluations w.r.t. tool reward.

| Method | $V^*$ | HR-4k | HR-8k |
|---|---|---|---|
| **w/o Tool Reward** | 87.4 | 53.4 | 55.4 |
| **Unconditional Reward** | 87.4 | 72.1 | 71.8 |
| **Conditional Reward** | 90.1 | 75.1 | 72.6 |

Table 6: **Scaling Model Size.** The 32B model is trained with the same data. Resp. Len.: Average Response Length. IoU is measured on $V^*$.

| Model | $V^*$ | WeMath | Resp. Len. | IoU |
|---|---|---|---|---|
| Qwen2.5-VL-7B | 71.2 | 34.6 | 212 | - |
| DeepEyes-7B | 90.1 | 38.9 | 241 | 0.37 |
| Qwen2.5-VL-32B | 87.9 | 47.7 | 314 | - |
| **DeepEyes-32B** | **93.3** | **55.9** | **754** | **0.53** |

Table 7: **Scaling Challenging Reasoning Data** from Chen et al. (2025) shows co-evolving perception ($V^*$) and mathematical problem-solving.

| Model | MVerse | WeMath | $V^*$ |
|---|---|---|---|
| Qwen2.5-VL-7B | 45.6 | 34.6 | 71.2 |
| DeepEyes-7B | 47.3 | 38.9 | 90.1 |
| **+ More Reasoning Data** | **51.8** | **43.6** | **91.6** |

Table 8: **Zero-Shot Tool Generalization.** HR-OCR-Rot: Random rotated subsets of HR-Bench-8K for OCR tasks.

| Model | $V^*$ | HR-OCR-Rot |
|---|---|---|
| Qwen2.5-VL-7B | 71.2 | 76.5 |
| DeepEyes (crop) | 90.1 | 80.1 |
| **DeepEyes (crop+rotate)** | **90.1** | **83.6** |

Table 9: **Ablation on iMCoT.** We provide results trained with text-only CoT on the same datasets.

| Model | $V^*$ | HR-4K | HR-8K |
|---|---|---|---|
| Qwen2.5-VL-7B | 71.2 | 68.8 | 65.3 |
| RL w. Text-only CoT | 88.5 | 75.4 | 60.8 |
| **DeepEyes (iMCoT)** | **90.1** | 75.1 | **72.6** |

• **Stage 3: Efficient Utilization (Steps 45–80)** The model adopts a more selective, precise approach, reducing query frequency and response length while maintaining high grounding and task accuracy. This reveals a compact visual-linguistic policy: active perception is invoked only when needed, complementing internal reasoning. High IoU with fewer queries reflects implicit planning, as the model narrows the visual scope internally before selectively confirming hypotheses.

Overall, training progresses from broad exploration to targeted exploitation, showing that the model can learn to integrate active perception into reasoning effectively. The ability to leverage active perception strategically co-evolves with its policy, highlighting the potential of perception-augmented visual-language models for scalable and interpretable multimodal reasoning.

**Tool Reward.** The reward in Eq. 2 includes a conditional component (*tool reward*) that grants a bonus only when the model answers correctly while performing active perceptions. For comparison, we train two variants: one without the conditional bonus (*w/o tool reward*) and one with an unconditional bonus (*unconditional reward*). Results are shown in Figure 4 and Table 5. Without the conditional reward, the model quickly reduces and stops performing perception actions. With an unconditional bonus, minimal engagement persists but remains static. Conditioning the reward on correctness leads to gradually increased active perceptions and more informative responses, reflecting deeper integration of visual reasoning. This setting achieves the highest accuracy, showing that rewarding actions alone are insufficient; alignment with correct outcomes is essential in *DeepEyes*.

**Thinking Patterns.** Here, we analyze diverse thinking patterns that emerged during end-to-end RL training, showing how the model performs active perceptions into its reasoning in ways that mirror human visual cognition. Four primary patterns can be identified: *1) Visual Search:* When facing complex problems that a single observation can't solve, the model actively scans different image regions, gathers visual clues, and reasons through them to reach reliable conclusions (Figure 7); *2) Visual Comparison:* When handling understanding across multiple images or objects, the model iteratively zooms in on each one, allowing close examination and comparison before drawing a final conclusion (Figure 8); *3) Visual Confirmation:* In some cases, the model begins with uncertainty but gradually builds confidence by zooming in on image details to gather evidence and resolve doubts (Figure 9); *4) Hallucination Mitigation:* Although VLMs can sometimes hallucinate, performing active perceptions helps the model focus on visual details to mitigate hallucination. (Figure 10).

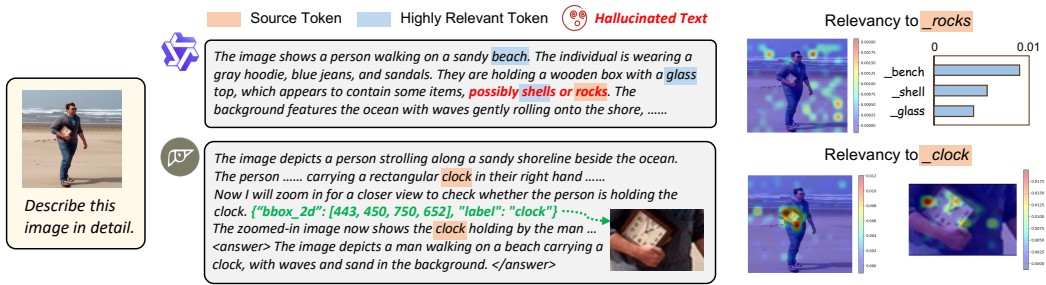

Figure 5: Analysis of hallucination mitigation. Qwen2.5-VL-7B (top) hallucinates "rocks," driven by linguistic association with "beach" rather than visual evidence, yielding low relevancy. In contrast, *DeepEyes* (bottom) triggers iMCoT to counter this bias, zooming in to re-ground reasoning and override the language prior, correctly identifying the "clock" with a focused relevancy heatmap.

## 4.4 ANALYSIS AND ABLATION STUDY

**Scaling Model Size.** Our framework exhibits strong scalability, as evidenced in Table 6. When scaling from 7B to 32B parameters, *DeepEyes* consistently widens its performance gap over the Qwen2.5-VL baseline. More importantly, the larger model demonstrates more sophisticated emergent behaviors. It generates substantially longer reasoning chains (Resp. Len.) and achieves higher grounding precision (IoU). This indicates that our RL paradigm not only boosts task performance but also fosters deeper and more accurate reasoning as model capacity increases.

**Scaling Challenging Reasoning Data.** As shown in Table 7, scaling our training set with more challenging reasoning data (from 23% to 42%) demonstrates a mutual reinforcement between perception and reasoning, improving performance on both mathematical benchmarks and the perception task $V^*$ as well. We hypothesize that stronger abstract reasoning enables a more sophisticated understanding of complex queries, which in turn guides a more effective visual-grounded thinking process.

**Zero-Shot Tool Generalization.** The primary goal of *DeepEyes* is to explore how models can natively "think with images," using cropping as a simple, foundational tool. Although not aimed at building a large toolset, the framework is easily extensible. To verify this, we introduced a *rotate* tool solely through the system prompt, requiring no retraining or architectural changes. We evaluated it on HR-OCR-Rot, a benchmark we created by applying random rotations $(0°, 90°, 180°, 270°)$ to the HRBench-8K OCR subset. As shown in Table 8, the tool yielded a 3.5% performance gain on this task while maintaining stable results on the general $V^*$ benchmark, demonstrating that *DeepEyes* can seamlessly integrate new tools and apply them selectively for zero-shot generalization.

**Ablation on iMCoT.** Finally, the ablation in Table 9 isolates the contribution of our core iMCoT mechanism. Compared to an RL baseline trained with a text-only CoT, iMCoT achieves superior performance across all benchmarks. The advantage is most pronounced on the ultra-high-resolution HR-8K benchmark, where iMCoT outperforms the text-only approach by a substantial margin. This result decisively demonstrates that for tasks requiring fine-grained visual detail, interleaving visual perception with textual reasoning is not merely beneficial, but essential for robust performance.

## 4.5 CASE STUDY: HOW DOES *DeepEyes* SYSTEMATICALLY MITIGATE HALLUCINATION?

Object hallucination in VLMs often stems from a strong language bias (Zhou et al., 2024), where text generation detaches from the visual input to rely on learned linguistic patterns. Our "thinking with images" paradigm directly counters this. By triggering active perception, the model is forced to re-engage with visual evidence, effectively fact-checking its linguistic assumptions against visual reality. To analyze this mechanism, we compute ***relevancy maps*** (Ben Melech Stan et al., 2024) to quantify the grounding of the model's output, which measures the contribution of all preceding tokens to the generation of a specific source token. Visualized via heatmaps, high relevancy attributed to image regions indicates strong visual grounding, whereas high relevancy from purely textual priors suggests a language-driven hallucination. As illustrated in Figure 5, this approach proves effective. While a baseline model succumbs to linguistic bias, *DeepEyes* leverages active perception to re-evaluate its initial assumptions based on new visual evidence. This process breaks ungrounded reasoning, overriding the language prior and correcting the hallucination, as confirmed by our relevancy analysis.

# 5 CONCLUSION

In this paper, we presented *DeepEyes*, a vision-language model that learns to "think with images" via end-to-end reinforcement learning. Unlike prior methods, this capability emerges natively, requiring neither pre-collected reasoning data for SFT nor external specialized models. To guide its reasoning behavior, we propose an active perception mechanism, featuring tailored data selection and rewards, that promotes successful reasoning trajectories by incentivizing the strategic use of visual grounding. Consequently, *DeepEyes* achieves competitive results on multiple benchmarks, exhibiting diverse, human-like reasoning patterns such as visual search and comparison.

## ACKNOWLEDGMENTS

This work was supported by the National Key Research and Development Program of China (2023YFB3107401), the National Natural Science Foundation of China (62521002, U24B20185). Thanks to the New Cornerstone Science Foundation and the Xplorer Prize.

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

## A PROMPT

### A.1 SYSTEM PROMPT

```
SYSTEM_PROMPT

You are a helpful assistant.

# Tools
You may call one or more functions to assist with the user query.
You are provided with function signatures within <tools></tools> XML
↪  tags:
<tools>
{
  "type": "function",
  "function": {
    "name": "image_zoom_in_tool",
    "description": "Zoom in on a specific region of an image by
    ↪  cropping it based on a bounding box (bbox) and an optional
    ↪  object label.",
    "parameters": {
      "type": "object",
      "properties": {
        "bbox_2d": {
          "type": "array",
          "items": {
            "type": "number"
          },
          "minItems": 4,
          "maxItems": 4,
          "description": "The bounding box of the region to zoom in,
          ↪  as [x1, y1, x2, y2], where (x1, y1) is the top-left
          ↪  corner and (x2, y2) is the bottom-right corner."
        },
        "label": {
          "type": "string",
          "description": "The name or label of the object in the
          ↪  specified bounding box (optional)."
        }
      },
      "required": [
        "bbox_2d"
```

```
        ]
      }
    }
  }
</tools>

# How to call a tool
Return a json object with function name and arguments within
↪  <tool_call></tool_call> XML tags:
<tool_call>
{"name": <function-name>, "arguments": <args-json-object>}
</tool_call>

**Example**:
<tool_call>
{"name": "image_zoom_in_tool", "arguments": {"bbox_2d": [10, 20,
↪  100, 200], "label": "the apple on the desk"}}
</tool_call>
```

## A.2 USER PROMPT

```
USER_PROMPT

Question: {}

Think first, call **image_zoom_in_tool** if needed, then answer.
↪  Format strictly as:  <think>...</think>
↪  <tool_call>...</tool_call> (if tools needed)
↪  <answer>...</answer>
```

# B TRAINING DATA

## B.1 DATA DISTRIBUTION

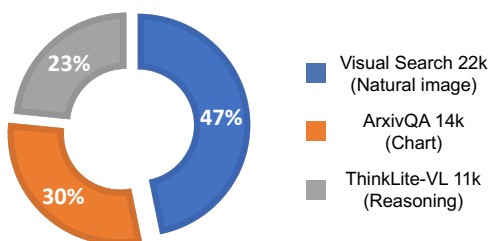

Figure 6: Distribution of Training Data.

As shown in Figure 6, our training corpus is constructed from three distinct sources, each contributing a unique focus:

- **Visual Search (47%, 22k samples)**: To support the model's visual grounding and fine-grained perception capabilities, we leverage the $V^*$ dataset (Wu & Xie, 2024), which is derived from COCO2017 (Lin et al., 2014). This collection emphasizes natural image understanding, where accurate responses require identifying subtle visual cues and object-level distinctions.
- **ArxivQA (30%, 14k samples)**: To diversify the visual input types, we incorporate the ArxivQA dataset (Li et al., 2024b), which features scientific plots, diagrams, and schematic charts. These samples introduce structured visual semantics beyond natural scenes, enabling the model to better interpret abstract and symbolic visual representations.

- **ThinkLite-VL (23%, 11k samples)**: While the above datasets cover visual understanding and diagram comprehension, they are limited in reasoning variety. To address this, we include multimodal question answering examples from ThinkLite-VL (Wang et al., 2025b), focusing on tasks such as arithmetic reasoning, commonsense inference, and problem solving. This addition is intended to improve general reasoning robustness and mitigate modality-specific overfitting.

## B.2 IMPACT OF TRAINING DATA

Table 10 reveals the critical role of training data composition. While unfiltered data (#1) offers minimal benefit, our curated, fine-grained data (#2) substantially boosts high-resolution image handling. However, this specialization induces catastrophic forgetting of reasoning skills. We address this by incorporating reasoning data (#3), which preserves mathematical abilities without sacrificing perception gains. To further enhance the model's cognitive range, we introduce chart data (#4), which adds visual diversity and fosters complex relational reasoning. The results confirm a clear synergy: high-resolution data for perception, reasoning data for cognitive retention, and chart data for relational complexity. Consequently, our final dataset (#5) combines these complementary sources to comprehensively activate the model's visual reasoning capabilities.

Table 10: **Impact of Training Data.** Fine represents the fine-grained data. HR denotes HR-Bench. Row #0 is the origin score of Qwen2.5-VL-7B.

| # | Fine | Reason | Chart | High-Resolution | | | Basic VL Capability | | Reasoning | |
|---|---|---|---|---|---|---|---|---|---|---|
| | | | | $V^*$ Bench | HR-4K | HR-8K | ReasonSeg | POPE | MathVista | MathVerse |
| **0** | | | | 71.2 | 68.8 | 65.3 | 68.3 | 85.9 | 68.2 | 45.6 |
| 1 | ✓ | | | 86.9 | 68.9 | 67.3 | 69.0 | 86.6 | 67.0 | 42.9 |
| **2** | ✓ | | | 91.6 | 74.1 | 71.0 | 69.1 | 88.1 | 64.7 | 41.3 |
| **3** | ✓ | ✓ | | 91.6 | 73.8 | 70.5 | 68.6 | 88.8 | 67.7 | 43.8 |
| **4** | ✓ | | ✓ | 90.1 | 74.6 | 74.6 | 68.5 | 87.9 | 64.6 | 38.1 |
| **5** | ✓ | ✓ | ✓ | 90.1 | 75.1 | 72.6 | 68.6 | 87.7 | 70.1 | 47.3 |

## C CO-FIRST AUTHOR CONTRIBUTIONS

- **Chenxiao**: Conducted early-stage exploration, contributed the main coding, and conducted the experiments.

- **Jack**: Conducted early-stage exploration and performed evaluation.

- **Michael**: Contributed codebase, and conducted the experiments and analysis.

- **Ziwei**: Performed data curation, completed the main manuscript writing, and conducted analysis.

## D MORE CASES

### D.1 SUCCESSFUL CASES

- **Visual Search**
  Figure 7: After an initial observation of the whole image, the model recognized that the current visual information alone was insufficient to determine whether it was wet, and acknowledged that factors such as lighting could cause misleading cues. It was therefore decided that a more detailed inspection was necessary. During the first tool invocation, grounding was inaccurate, and the cropped image failed to provide some clues. The model then conducted a second grounding step, this time actively focusing on the area surrounding the wetsuit in an attempt to locate more direct indicators—such as water droplets or visible signs of wetness. It also incorporated contextual cues from the surrounding environment, such as reflections on wet sand and the wetsuit's contact with water. Ultimately, by combining zoomed-in visual details—such as the wetsuit's dark coloration and how it clung to the body—with indirect environmental evidence, the model concluded that the wetsuit appeared to be wet.

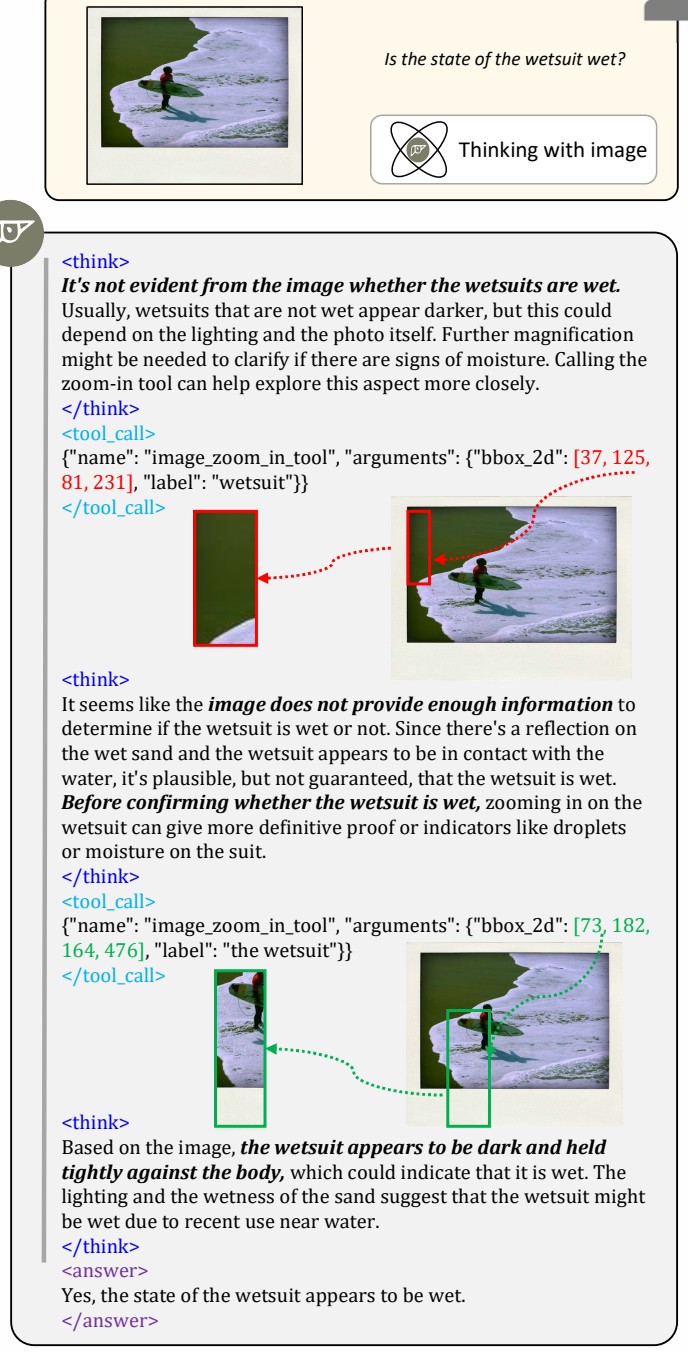

Figure 7: Thinking Pattern: Visual Search.

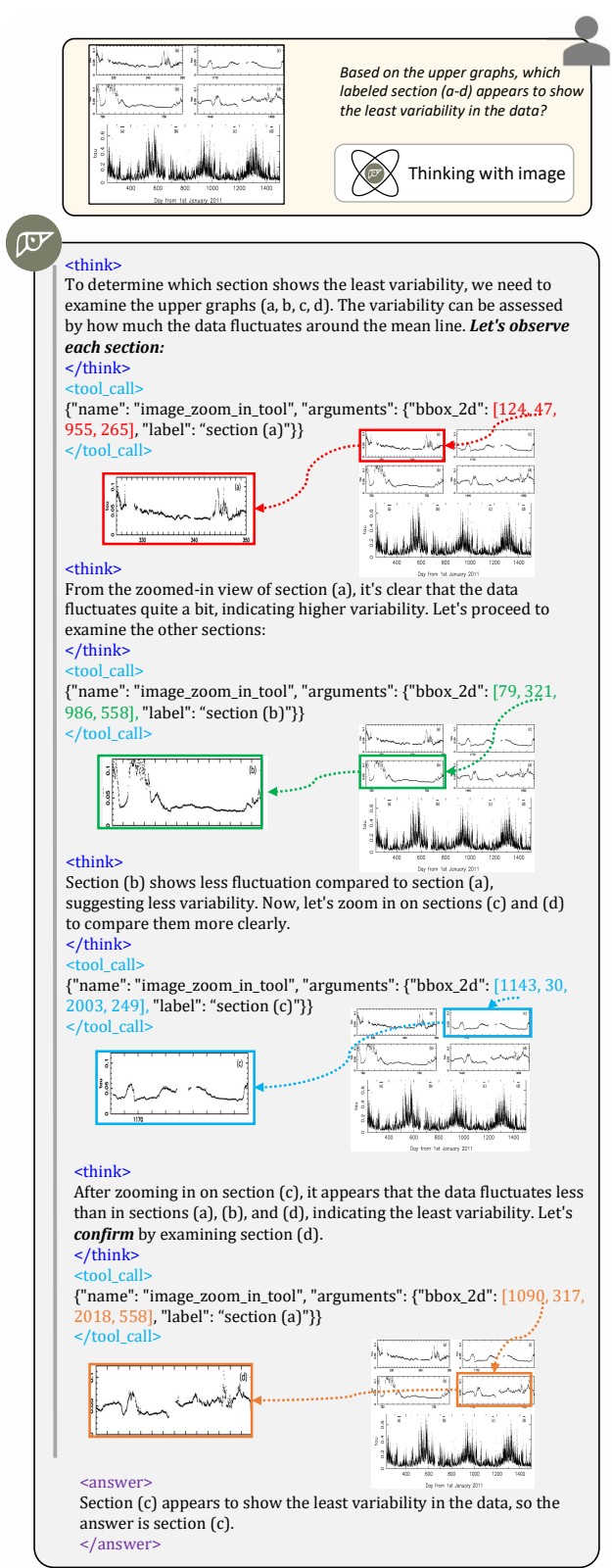

Figure 8: Thinking Pattern: Visual Comparison.

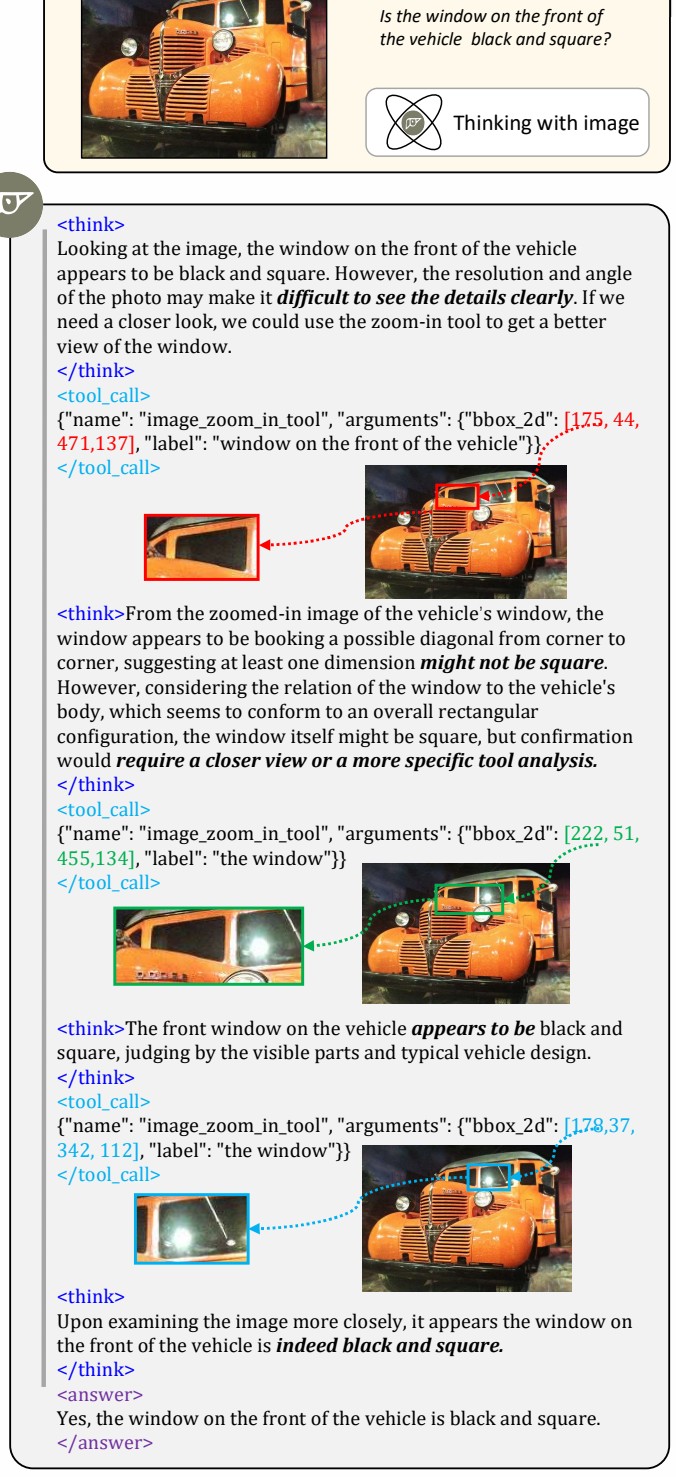

Figure 9: Thinking Pattern: Visual Confirmation.

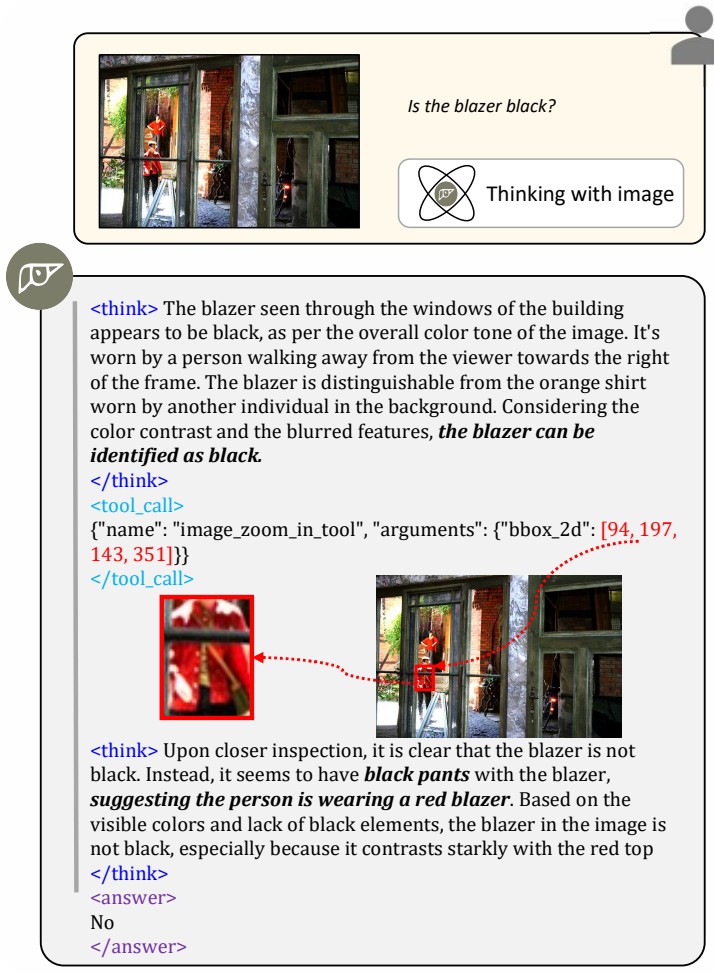

Figure 10: Thinking Pattern: Hallucination Mitigation.

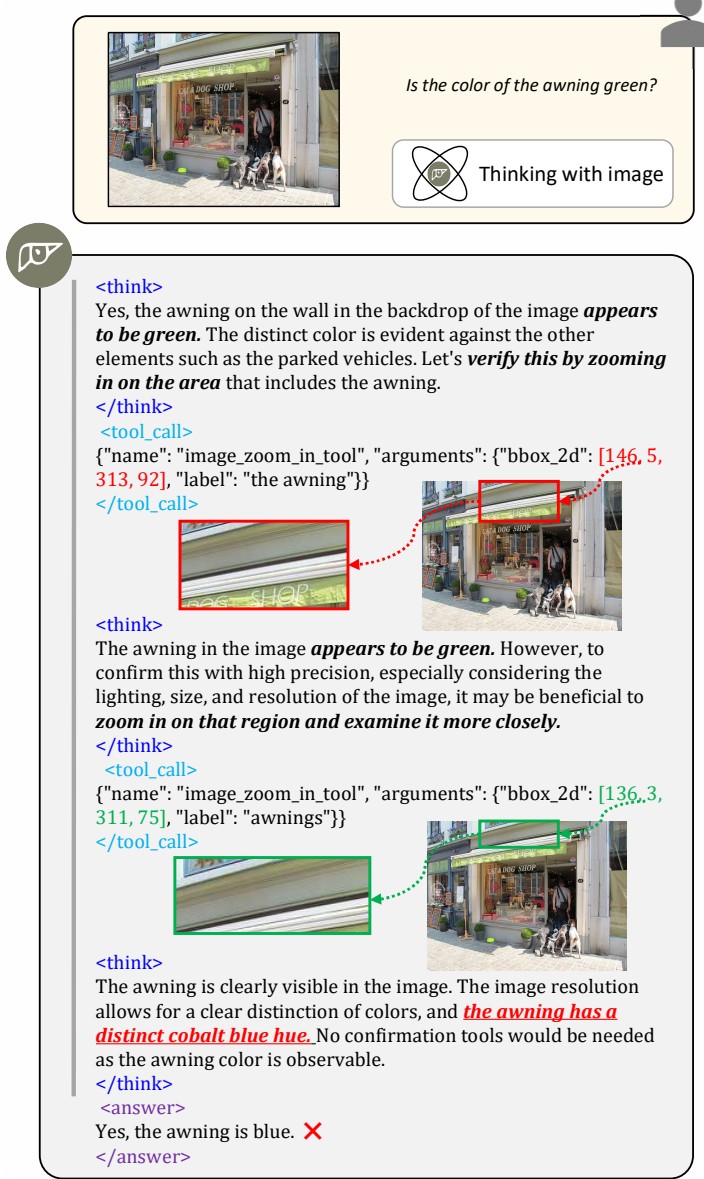

Figure 11: Grounding Limitation.

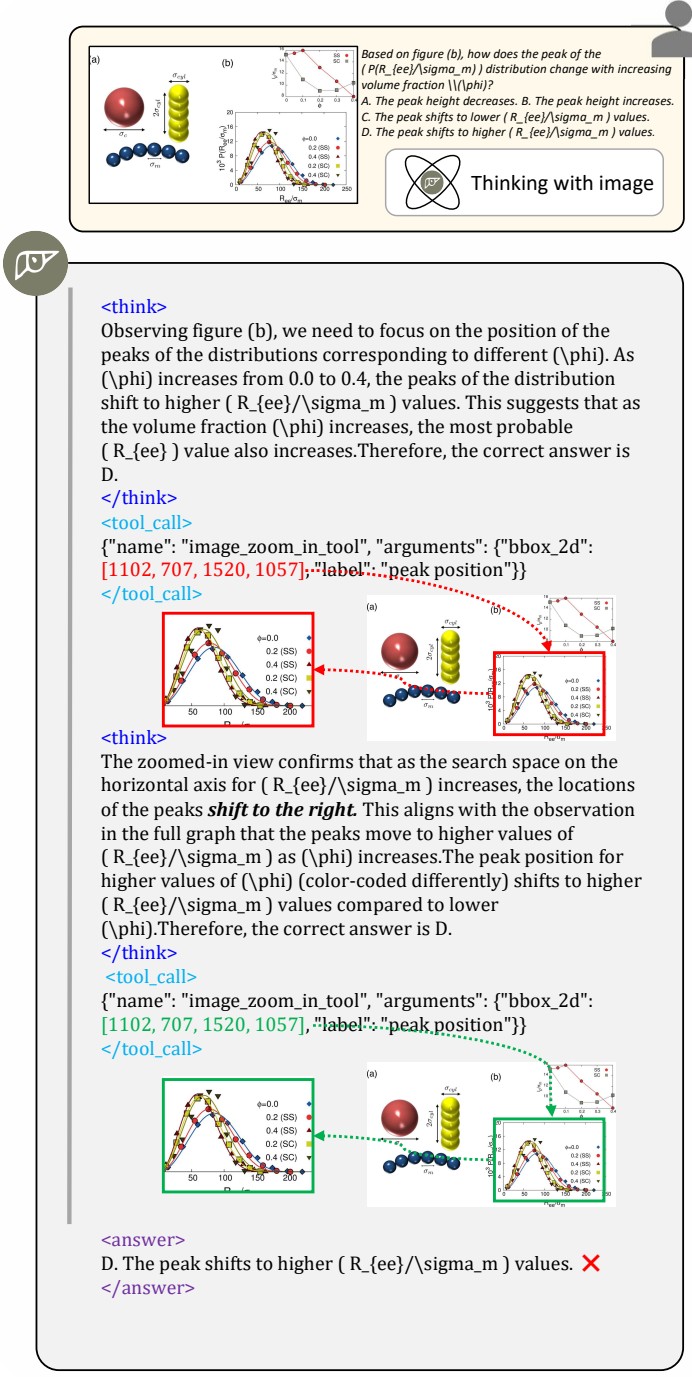

Figure 12: Reasoning Limitation.

- **Visual Comparison**

  Figure 8: To determine which section exhibits the least data variability, the model sequentially zoomed in on the charts of four sections (a, b, c, and d), focusing on fluctuations around the moving average. Through comparison, it found that section (a) showed significant volatility, while section (b) was relatively less volatile. However, section (c) displayed the most stable pattern, with fluctuations clearly smaller than those in the other regions. Based on this analysis, the model concluded that section (c) has the least data variability.

- **Visual Confirmation**

  Figure 9: In this case, the model was initially uncertain about the shape of the window. Through multiple invocations of the zoom-in tool and careful analysis of potential visual details, it gradually resolved its internal uncertainty and ultimately provided a confident answer.

- **Hallucination Mitigation** Figure 10: The model initially confused the colors of the pants and the blazer. However, by leveraging its perceptual capabilities and invoking the zoom-in tool to examine the enlarged region, it ultimately corrected the hallucination.

### D.2 FAILED CASES

- **Grounding Limitation**

  Figure 11: The model initially hypothesized that the awning was green. It then invoked the zoom-in tool for a closer inspection, maintaining its assumption while noting the need for more precise verification. However, during the second zoom-in, grounding drift occurred—the awning was no longer within the selected region, and instead, a blue area appeared. This misalignment led to a reversal in the model's judgment, ultimately resulting in an incorrect answer.

- **Reasoning Limitation**

  Figure 12: Although the model was able to accurately locate the position of figure (b) and invoke the tool for detailed inspection, it still lacked fine-grained understanding and reasoning capabilities. It failed to thoroughly analyze the trend changes in the zoomed-in curves, ultimately leading to an incorrect answer.

## E LIMITATIONS

Although the simple end-to-end RL can elicit visual reasoning abilities, there still exist shortcuts, such as insufficient richness in the reasoning process and inaccurate target localization. We think these issues stem from limitations in the foundation model's poor capabilities. We only utilized Qwen2.5-VL-7b, which has relatively weak fundamental capabilities due to its small model size.

## F BROADER IMPACTS

Our exploration of interleaved multimodal chain-of-thought reasoning provides valuable insights for the future development of the AI community. By investigating how models can engage in step-by-step visual reasoning through interactive dialogues, we advance understanding of more transparent and interpretable AI systems. This research direction may inspire new architectures and training methodologies that better align with human reasoning processes.

## G FUTURE WORK

Currently, our visual reasoning process only includes the crop operation. However, in real-world scenarios, a wider range of tools is needed, such as search and drawing auxiliary lines. We will explore the integration of additional tool utilization in our future work.

