# OpenReview forum: "DeepEyes: Incentivizing "Thinking with Images" via Reinforcement Learning"
_ICLR.cc/2026/Conference — ICLR 2026 Poster_

### Official Review · Reviewer_gPB1 · 2025-10-26

**Soundness:** 3
**Presentation:** 4
**Contribution:** 3
**Rating:** 8
**Confidence:** 3

**Summary:**

This paper presents DeepEyes as one of the first open-source solutions that follow the thinking-with-images paradigm and uses image cropping as a tool to incentivize fine-grained visual perception capabilities in modem VLMs.

Two key contributions of this paper are:
- Interleaved Multi-modal Chain-of-Thought (iMCoT) that reasons beyond text-only trajectories and provides seamless integration of zoom-in operations and textual reasoning.
- Conditional tool reward that promotes the usage of zoom-in actions.

While those key contributions are common practice in agentic literature, I think it is not a trivial attempt to make them work in the thinking-with-images paradigm. Considering the comprehensiveness of ablation studies in this paper, I lean to recommend acceptance pending the discussion with fellow reviewers/ACs.

**Strengths:**

1. DeepEyes comes with clear motivation – using RLVR-style post-training methods to activate the precise grounding capabilities of VLMs.
2. The proposed iMCoT and conditional tool reward prove to be effective in ablation studies in various high-res & visual reasoning benchmarks.
3. The staged evolution (exploration -> high-frequency engagement -> efficient utilization) is compelling and supported with metrics like tool-call counts, response length, and grounding IoU curves.

**Weaknesses:**

1. Both iMCoT and conditional tool incentives are now fairly standard in agentic RL literature (interleaving actions + CoT, rewarding tool use). The novelty is mainly in making them work cleanly for “thinking with images”.
2. While the authors claim to put DeepEyes in an agentic framework, the available tools are relatively limited – seems to be zoom-ins only. See the following Q1.

**Questions:**

1. Can you anticipate the generalization beyond image crop? Say various image manipulation tools in related work, such as scale, contrast, denoise, or keypoint heatmaps.
2. Can you quantify the benefit of (i) difficulty curation, (ii) verification, and (iii) perception-utility filtering in your data curation pipeline?

---

> ### Author Response · Authors · 2025-11-22
> **Response to Reviewer gPB1**
>
> > ## **W1: Novelty in making work for "thinking with images".**
>
> We agree with this assessment. Our core contribution is formulating "Active Perception" rather than inventing new RL algorithms. We represent a pioneering effort to utilize end-to-end RL to incentivize this mechanism and validate it across general perception and reasoning benchmarks.
>
> Thus, our novelty stems from the Active Perception mechanism itself, distinct from the standard RL techniques employed. By making this work "cleanly," as the reviewer notes, we provide the proof-of-concept that VLMs can internalize active perception as an intrinsic cognitive step, moving beyond reliance on external APIs.
>
> ---
>
> > ## **W2/Q1: Limited Tools.**
>
> We acknowledge the reviewer's observation that our current system heavily utilizes cropping. However, the framework is fundamentally tool-agnostic, designed to treat any visual operation as an internal reasoning step rather than being limited to zoom-ins. To empirically demonstrate this generalization beyond the "rotate" tool already shown in Table 8, we integrated a "flip" tool without any retraining and evaluated it on a constructed HRBench-OCR-flip dataset. The results confirm our system's flexibility: DeepEyes (crop+flip) effectively utilizes the new tool to handle mirrored text, achieving 81.1% accuracy compared to the baseline Qwen2.5-VL's 76.5%.
>
> | Model                  |  V*  |  HRBench-OCR-flip  |
> | ---------------------- | :--: | :----------------: |
> | Qwen2.5-VL-7B          | 71.2 |        76.5        |
> | DeepEyes (crop)        | 90.1 |        80.6        |
> | DeepEyes (crop+flip)   | 89.0 |        81.1        |
>
> Crucially, this experiment also validates our design choice to prioritize cropping. The data reveals that the primary performance leap stems from the "crop" mechanism, while the specialized "flip" tool provides a marginal, albeit useful, supplementary gain. This reinforces our premise that "active zooming" (Crop) is the most fundamental and high-impact operator for resolving general visual ambiguity, while other geometric transformations serve more niche scenarios.
>
> Furthermore, as for other operations, we view these as exciting directions for future work, now that the core reasoning mechanism for active perception has been successfully established.
>
> ---
>
> > ## **Q2: Ablations on data curation pipeline.**
>
> We appreciate the suggestion to quantify the specific contribution of each stage in our data curation pipeline. To isolate these benefits, we conducted a progressive ablation study on the V* benchmark. The results are summarized in the table below:
>
> | Strategy             | V*   |
> | -------------------- | ---- |
> | w/o Filter (Full)    | 86.4 |
> | + Difficulty         | 86.9 |
> | + Verification       | 87.4 |
> | + Perception-utility | 90.1 |
>
> The baseline model trained on unfiltered data achieves an accuracy of 86.4%. Integrating difficulty curation improves performance to 86.9% by eliminating samples that are statistically too trivial or impossible to learn. The subsequent data verification step contributes an additional 0.5% gain by removing samples with incorrect ground-truth labels. Most significantly, the perception-utility filter yields a substantial improvement of 2.7% and brings the final accuracy to 90.1%. This pronounced jump validates that retaining samples specifically solvable via active perception is the primary driver of the observed performance gains, while the preceding steps ensure basic data hygiene.

---

> > ### Comment · Reviewer_gPB1 · 2025-11-24
> >
> > Thank you for the additional ablation studies! I believe the current rating is a fair evaluation to your work. I keep acceptance recommendation.

---

### Official Review · Reviewer_BabE · 2025-10-29

**Soundness:** 3
**Presentation:** 3
**Contribution:** 3
**Rating:** 6
**Confidence:** 4

**Summary:**

The paper introduces DeepEyes, a vision-language model trained end-to-end with reinforcement learning to “think with images” by interleaving textual chain-of-thought with active visual perception. The model autonomously decides when to zoom into image regions by generating bounding boxes; the resulting crops are fed back as observation tokens and used for subsequent reasoning. Training uses agentic reinforcement learning with GRPO, sparse outcome rewards, and a conditional bonus that incentivizes correct answers achieved via active perception. A data curation pipeline selects samples likely to benefit from grounding-assisted reasoning. DeepEyes achieves strong gains on fine-grained, high-resolution benchmarks, improves general perception and reasoning, reduces hallucination and slightly improves grounding, and yields consistent math reasoning gains. The paper analyzes training dynamics, showing a three-stage evolution from exploration to efficient use of active perception, validates the conditional tool reward, demonstrates scalability to larger models, positive data scaling effects, and zero-shot tool extension.

**Strengths:**

- Originality:
  - Proposes an interleaved multimodal CoT (iMCoT) that natively integrates active perception into reasoning with end-to-end RL, avoiding pre-collected reasoning SFT and external specialized models/APIs (Abstract; Sections 1, 3.1).
  - Conditional reward that only bonuses correct tool-using trajectories effectively incentivizes perception-aware reasoning while discouraging gratuitous tool calls (Section 3.2; Table 5; Figure 4).
  - A targeted data curation pipeline that filters for perception-utility to bootstrap active perception without SFT (Section 3.3; Appendix B; Table 10).
- Quality:
  - Comprehensive evaluations across high-resolution, grounding/hallucination, and reasoning benchmarks with strong improvements (Tables 1–4).
  - Ablations isolate key components: conditional tool reward (Table 5), iMCoT vs text-only RL (Table 9), data composition (Table 10), model scaling (Table 6), and zero-shot tool generalization (Table 8).
  - Insightful training-dynamics analysis revealing emergent stages of active perception and behavioral patterns (Figure 3; Section 4.3), plus qualitative case studies (Figures 7–10).
- Clarity:
  - Clear agentic RL formulation with observation tokens (Section 3.2), system/tool interface (Appendix A), and pipeline overview (Figure 2).
  - Training details provided (Section 4.1) and code link for reproducibility.
- Significance:
  - Large, consistent gains on fine-grained high-res understanding (Table 1) and reductions in hallucination (Table 3), suggesting a practical path to multimodal test-time scaling via native visual thinking.
  - Demonstrates scalability to larger models with stronger grounding IoU and longer reasoning chains (Table 6), and extensibility to new tools at inference (Table 8).

**Weaknesses:**

- Technical detail gaps:
  - Reward magnitudes/normalization, exact coefficients for accuracy/format/tool rewards, and sensitivity analyses are not fully specified (Section 3.2; Table 5 references but no hyperparameters), limiting reproducibility and insight into stability.
  - Limited description of how image crops are tokenized/encoded, how many visual tokens per crop, and how observation tokens are interleaved with text for different tools (Section 3.2 mentions loss masks but not encoder specifics).
- Evaluation scope and fairness:
  - High-resolution gains are compelling, but comparisons to workflow baselines may be confounded by different training data and compute; stronger apples-to-apples controls are desirable (Table 1). Statistical significance and variance are not reported.
  - Grounding improvements on refCOCO family are modest (+0.6–1.0; Table 3); ReasonSeg gains are small. It would help to quantify how often crops improve IoU versus distract.
- Generality of “thinking with images”:
  - The system currently centers on cropping; while rotate shows zero-shot extensibility (Table 8), the toolset remains narrow. Some failure cases highlight grounding drift and reasoning limits (Figures 11–12), suggesting brittleness for complex visual workflows.
- Compute and efficiency:
  - Training uses 80 iterations on H100 with 256 prompts × 16 rollouts and max 6 perceptions, KL=0 (Section 4.1). Sample efficiency and cost-benefit vs SFT or hybrid methods are not deeply analyzed; KL=0 may risk policy drift/collapse.
- Claims around “no cold-start SFT”:
  - Although no reasoning SFT is used, the base model (Qwen2.5-VL) is already instruction-tuned; clarifying the exact initialization and any prompt engineering dependencies (Appendix A) would temper the claim.

**Questions:**

- Reward design and stability:
  - What are the exact numerical weights for R_acc, R_format, and the conditional R_tool? How sensitive are results to these weights and to the correctness indicator threshold (Eq. 2)? Please provide a sensitivity study or at least ranges (Section 3.2; Table 5).
  - Why was KL set to 0? Did you observe policy collapse or reward hacking at any point? Would a small KL or reference model improve stability/generalization (Section 4.1)?
- Observation/crop handling:
  - How are cropped images preprocessed and encoded (e.g., AnyRes vs fixed resolution), and how many visual tokens do they add? Is there an explicit cap on total image tokens across multiple crops? Any latency analysis (Section 3.1)?
  - Do you condition bounding-box generation on a learned coordinate head or pure text token decoding? How is the bbox format enforced and validated during rollouts (Appendix A)?
- Data and splits:
  - For V* and HR-Bench, please detail train/val/test splits used for RL vs evaluation to rule out leakage (Sections 3.3, 4.1). Are any evaluation images seen during RL?
  - Can you release the curated indices and the perception-utility labels (Section 3.3; Table 10)?
- Comparisons and metrics:
  - For Table 1, can you report confidence intervals or variance over multiple seeds? Similarly, could you add a compute-normalized comparison against Pixel-Reasoner (Su et al., 2025) and ZoomEye (Shen et al., 2024a)?
  - For grounding, can you report IoU distributions and success vs failure breakdown for cases where iMCoT invoked cropping vs not, to substantiate causal benefits (Section 4.2; Table 3)?
- Generalization and tools:
  - Beyond rotate, have you tried text-conditioned detection/segmentation or simple measuring tools (e.g., rulers/lines)? Any no-retrain extensions besides rotation (Table 8)?
  - What are the failure modes that lead to grounding drift (Figure 11)? Could a recurrent state or memory help prevent drift across sequential crops?

---

> ### Author Response · Authors · 2025-11-22
> **Response to Reviewer BabE - 1 / 3**
>
> > ## **W1/Q1-3: Technical details.**
>
> ---
>
> * **W1/Q1: Reward design.**
>
> Thank you for this suggestion. We formulate the coefficients for each reward as: $\alpha$ for $R_{acc}$, $\beta$ for $R_{format}$, and $\gamma$ for $R_{tool}$ in Eq.2. Results of the ablation study are shown below.
>
> | $\alpha$ | $\beta$ | $\gamma$ |     V* acc. (%)     |
> | :--------: | :-------: | :--------: | :-----------------: |
> |    1.0    |    0.0    |    1.2    | *format collapse* |
> |    0.8    |    0.2    |    0.4    | 86.9 |
> |    0.8    |    0.2    |    2.0    | 87.4 |
> |    **0.8**    |    **0.2**    |    **1.2**    | **90.1** |
>
> As shown in the first row, removing the format reward caused *format collapse*, confirming the necessity of a small explicit signal ($\beta$=0.2) to enforce structural validity without overshadowing the primary objective, consistent with standard agentic RL practices.
>
> However, the tool reward $\gamma$ was important; we found $\gamma$=1.2 to be optimal. More critical than the exact value was the ***conditional*** nature of the tool reward (ensuring a correct answer), which, as shown in our original ablation (Table 5), is essential for stable learning and preventing reward hacking. We will add these specific coefficients and the sensitivity analysis to Sec. 3.2.
>
> ---
>
> * **Q2: Observation/crop handling.**
>
> **Box Generation & Validation**: We use pure text token decoding. The model is conditioned via the system prompt (Appendix A) to generate a textual JSON call (e.g., {"name": "image_zoom_in_tool", "arguments": ...}). This text is then parsed and validated by our tool-execution framework during rollouts. We do not use a separate learned coordinate head.
>
> **Crop Preprocessing & Encoding**: Cropped images are preprocessed and encoded using the native vision encoder of the Qwen2.5-VL model, which supports flexible resolutions (akin to AnyRes).
>
> **Token Count & Cap**: The resulting observation tokens are appended to the trajectory. On $V^*$, this adds an average of 190 extra image tokens per query. We do not set an explicit cap on the total image tokens from multiple crops, but we limit the maximum number of perception steps to 6 per rollout, with the max response length (20480 tokens) acting as an implicit cap. A token-wise loss mask is applied to these non-model-generated observation tokens.
>
> **Latency Analysis**: The latency and token overhead on $V^*$ is as follows and tested on H20 GPUs. While the extra visual tokens introduced by iMCoT, the associated additional latency is ***negligible*** (~1.3%) compared to the total inference time. Thus, invoking the vision encoder with multiple times does not create a bottleneck. Besides, in real-world deployments, this overhead is minimized by modern serving infrastructure. Techniques such as KV Caching, Encode-Prefill-Decode (EPD) Disaggregation [1,2] can efficiently batch the processing of new image crops during the prefill phase.
>
> | Model       | Avg. Total Tokens | Total Latency | Extra Image Tokens | Extra Latency |
> | ----------- | :---------------: | :-----------: | :----------------: | :------------: |
> | DeepEyes-7B |        431        |      39s      |        190        | 51.5ms (~1.3%) |
>
> Furthermore, our training dynamics (Fig. 3, Stage 3) show the model ***learns to be efficient***. It learns to selectively invoke this scaling only when necessary, converging on a policy that balances accuracy and cost.
>
> ---
>
> * **Q3: Data and splits.**
>
> Our RL training uses the official training splits from V* (derived from COCO2017 train), ArxivQA, and ThinkLite-VL. All evaluations (Sec. 4.1) are performed exclusively on the official test sets of the benchmarks. For benchmarks with explicit train/test splits (like V*), we train on the train set and evaluate on the test set. For most other benchmarks, they serve as ***zero-shot, out-of-domain*** evaluations as their data was not seen during training. We will also release our curated dataset indices and the perception-utility labels upon publication to ensure full reproducibility.
>
> ---
>
> ### **Reference**
>
> [1] ModServe: Modality- and Stage-Aware Resource Disaggregation for Scalable Multimodal Model Serving. arXiv:2502.00937.
>
> [2] HydraInfer: Hybrid Disaggregated Scheduling for Multimodal Large Language Model Serving. arXiv:2505.12658.

---

> ### Author Response · Authors · 2025-11-22
> **Response to Reviewer BabE - 2 / 3**
>
> > ## **W2: Evaluation scope and fairness.**
>
> ---
>
> * **Comparison to workflows.**
>
> We believe that a perfectly controlled data comparison is challenging, as the training strategy is intrinsically tied to the method (e.g., specialized SFT data for workflows vs. our RL data curation). To provide a more direct, base-model-controlled comparison, we refer to the MME-RealWorld-Lite benchmark (Tab. 2). While workflow methods like ZoomEye did not report scores on our Qwen2.5-VL-7B base in their paper, results are available on public leaderboards[3]. This allows a clearer comparison:
>
> | Model         |     Type     | MME-RealWorld-Lite |
> | ------------- | :-----------: | :----------------: |
> | Qwen2.5-VL-7B |  Base Model  |        42.3        |
> | ZoomEye       |   Workflow   |        48.1        |
> | DeepEyes      | End-to-End RL |        53.2        |
>
> This result (+10.9 vs. base, +5.1 vs. workflow) strongly supports that our RL-based iMCoT provides a more significant improvement over both the base model and complex, manually-defined pipelines.
>
> ---
>
> * **Causal benefits of cropping (IoU vs. Distraction).**
>
> We thank the reviewer for this point. We first clarify that the benchmarks in Tab.3 (e.g., refCOCO) evaluate ***pure grounding ability*** with IoU-based metrics. The results show this fundamental ability improves slightly, even without an explicit IoU reward, solely from end-to-end RL on downstream tasks. More directly, our analysis in Sec.4.3 tracks the IoU of the actual crops generated during reasoning. We observe a strong implicit link: a precise crop (high IoU) provides useful information, leading to a correct final answer and high benchmark scores. Conversely, an inaccurate crop (low IoU) serves as a distraction, leading to an incorrect answer. Therefore, the significant improvement in the average IoU of taken crops (Fig.3c) is not just a measure of grounding; it is a direct proxy for the utility of the perception action. The model learns to crop more effectively precisely because higher-IoU crops are causally linked to achieving higher task accuracy. ***This analysis already shows the model learns to favor beneficial crops over distracting ones.***
>
> ---
>
> > ## **W3: Generality of “thinking with images”.**
>
> We focus on **Active Perception through Reasoning** and select "crop" as the primary operation because it essentially models the cognitive act of "looking closer." Unlike agents that use tools as external APIs, DeepEyes internalizes visual operations as reasoning steps. As validated by the zero-shot extension to "rotate," the framework is generalizable and ready to support broader toolsets.
>
> ---
>
> > ## **W4: Compute and efficiency.**
>
> ---
>
> * **Sample efficiency vs SFT.**
>
> While our RL training requires significant compute (80 iterations, 256 prompts x 16 rollouts), it bypasses the need for ***large-scale, costly annotation of intermediate reasoning steps, which is a prerequisite for SFT-based models.*** RL shifts the cost from manual offline data curation to automated online computation, which we argue is a more ***scalable*** path for developing complex reasoning abilities.
>
> ---
>
> * **Why KL=0.**
>
> We set KL=0 primarily because our iMCoT reasoning process, which involves interleaved text and visual tool calls, represents a significant ***policy shift*** from the base model's standard text generation. We hypothesized that a strict KL constraint against the base model might overly restrict the RL search space and prevent the model from discovering this new, complex agentic behavior. This approach of removing the KL constraint to allow for greater policy exploration is an increasingly ***common practice*** in RLVR for complex reasoning[4] and agentic[5] tasks. Our results show this strategy was effective. The policy did not collapse, and we avoided reward hacking without a KL penalty.
>
> ---
>
> > ## **W5: Claims around “no cold-start SFT”.**
>
> Our claim "no cold-start SFT" specifically means we do not use any pre-collected SFT data on ***intermediate reasoning steps or tool-use trajectories.*** This contrasts with prior methods that often require supervised data to "teach" the model how to use tools or follow a workflow. We start from the standard, publicly available Qwen2.5-VL-7B model, which is ***instruction-tuned on general tasks.*** Our contribution is demonstrating that the complex, interleaved iMCoT behavior can emerge from this general-purpose model, guided only by end-to-end RL with outcome-based rewards.
>
> ---
>
> ### **Reference**
>
> [3] https://huggingface.co/spaces/omlab/open-agent-leaderboard.
>
> [4] DAPO: An Open-Source LLM Reinforcement Learning System at Scale. arXiv:2503.14476.
>
> [5] Mobile-Agent-v3: Fundamental Agents for GUI Automation. arXiv:2508.15144.

---

> ### Author Response · Authors · 2025-11-22
> **Response to Reviewer BabE - 3 / 3**
>
> > ## **Q4: Comparisons and metrics.**
>
> ---
>
> * **Multiple seeds and compute-normalized comparison.**
>
> We have updated our results by running 3 independent seeds, yielding a mean accuracy of **90.37 ± 0.74 on $V^*$**, which confirms the robustness of our method. To provide a compute-normalized comparison with Pixel-Reasoner, we constrained the input image tokens (w/ pixel constraints to {640, 1280, 2560}×28×28) to match their settings. As shown in the table below, at similar average image token consumption, DeepEyes consistently outperforms Pixel-Reasoner on $V^*$.
>
> | Input pixel const. | Pixel-Reasoner ($V^*$) | Pixel-Reasoner (Token Count) | DeepEyes ($V^*$) | DeepEyes (Token Count) |
> | :----------------- | :----------------------: | :--------------------------: | :----------------: | :--------------------: |
> | 640×28×28        |           64.4           |             680             |   **64.8**   |          685          |
> | 1280×28×28       |           74.3           |             1563             |   **75.9**   |          1374          |
> | 2560×28×28       |           74.2           |             2355             |   **80.0**   |          2605          |
>
> ---
>
> * **Grounding metrics.**
>
> See our reply to W2.
>
> ---
>
> > ## **Q5: Generalization and tools.**
>
> ---
>
> * **Other no-retrain extensions.**
>
> To further demonstrate zero-shot extensibility, we integrated a **"flip"** tool. Using a protocol similar to rotation, we constructed HRBench-OCR-flip. As shown below, the flip tool effectively handles mirrored text, improving performance to 81.1%. Crucially, the comparison reveals that the primary performance leap comes from the "crop" mechanism (76.5% $\rightarrow$ 80.6%), while "flip" provides a marginal supplementary gain (80.6% $\rightarrow$ 81.1%). This result reinforces our paper's premise: that "active zooming" (cropping) is the most fundamental and high-impact operation for resolving visual ambiguity in general perception, justifying our system's primary focus on this mechanism.
>
> | Model                  |  V*  |  HRBench-OCR-flip  |
> | ---------------------- | :--: | :----------------: |
> | Qwen2.5-VL-7B          | 71.2 |        76.5        |
> | DeepEyes (crop)        | 90.1 |        80.6        |
> | DeepEyes (crop+flip)   | 89.0 |        81.1        |
>
> ---
>
> * **Grounding drift.**
>
> This is an excellent observation and a key challenge. The failure case in Fig.11 illustrates this drift, where the second zoom-in loses the target "awning" and focuses on a "blue area" instead, leading to an incorrect answer. We believe this is due to the stateless nature of the tool call. Your suggestion of using a recurrent state or memory (e.g., by feeding the previous BBox coordinates back into the context) is a very promising direction to help the model maintain context across sequential perceptions. We thank you for this suggestion and will explore it in future work.

---

### Official Review · Reviewer_3FYG · 2025-10-31

**Soundness:** 3
**Presentation:** 3
**Contribution:** 4
**Rating:** 6
**Confidence:** 5

**Summary:**

This paper proposes a new CoT pipeline by integrating both the text and visual cues into the intermediate reasoning steps. Specifically, without SFT training, this paper firstly curates a post-training dataset and then leverages the RL-like GRPO algorithm to incentivize the model's internal grounding abilities to crop and embed the local visual information into the next reasoning trajectory. Trained by several reward functions, this paper then achieves powerful performances on general perception tasks and also the reasoning benchmarks, which imitates our human beings' zoom-in ability to investigate the mostly related visual regions and does not solely rely on once visual embedding sequencers and text-based reasoning trajectories.

**Strengths:**

1. Combing both the textual and visual cues into the MLLM's intermediate reasoning trajectories sounds like an interesting and critical exploration, despite the openai-o3 has applied the very similar methods to empower the MLLM thinking with images abilities.

2. The fully released pipeline including codes and datas contributes the community which can be a good point for the community to develop the O3 frameworks.

3. The overall pipeline is not that complicated and somehow simple but effective, in which the authors also conduct various experiments to explore and exploit the overall RL training dynamics.

**Weaknesses:**

1. Regarding the iterative MCoT steps, does this paper explore multiple object grounding at the same time?

2. If the model needs more than twice MCoT, then the model needs more than twice visual embedding extraction, which sounds like a computational overhead. Can the authors also provide more inference latency analysis?

3. What if the model predicts all possible grounding objects' bounding boxes and extracts all visual embeddings, then concatenates all these new local visual tokens together, saying all in once?

4. What about the hard image-text samples, but with a small resolution? Does the author study these low-resolution scenarios?

5. I also notice that the implemented codes adopted a stronger MLLM models using vllm to serve as the judger, but the paper claims internal grounding post-training, which raises confusions.

**Questions:**

I am also curious what if using these thinking with images idea into video tasks or 3D scenarios, by adopting and improving the similar pipeline into other more difficult tasks?

**Details Of Ethics Concerns:**

No.

---

> ### Author Response · Authors · 2025-11-22
> **Response to Reviewer 3FYG - 1 / 2**
>
> > ## **W1: Multiple object grounding.**
>
> Thank you for this question, which clarifies a key aspect of our iMCoT mechanism. Our current approach is ***iterative and sequential***. In each active perception step, the model grounds and crops a single region of interest (e.g., Fig. 8, where regions (a)-(d) are examined sequentially). This 'focus-and-shift' process mimics human attention and allows reasoning from one step to guide the perception of the next.
>
> From an information gain perspective, simultaneous (parallel) and sequential grounding are largely equivalent. Specifically, attending to distinct objects across a short reasoning trajectory can be regarded as a form of multi-object grounding. The key difference lies in the insertion point of the new visual tokens into the CoT. Our work prioritizes incentivizing a robust step-by-step reasoning process.
>
> We did explore simultaneous, multi-region grounding in early experiments. However, this approach proved challenging, as it placed a significantly higher demand on the model's ***multi-image handling capacity in the same turn*** and introduced formatting control issues, which often led to training instability. We therefore adopted the sequential method for its stability and alignment with our core objective. We consider exploring simultaneous grounding on more capable foundation models as a promising direction for future work.
>
> ---
>
> > ## **W2: Computational overhead of visual embedding extraction.**
>
> Thank you for this crucial point. You are correct that iterative iMCoT steps introduce computational overhead, as each cropped image (e.g., $I_{t1}$) requires a separate pass through the vision encoder (Fig. 2). Our goal is to solve complex, fine-grained tasks (e.g., +18.9% on V*) where standard single-pass models fail. We view this extra computation as a form of ***test-time scaling***. To quantify this, we provide a supplementary analysis of DeepEyes-7B on the V* benchmark with H20 GPU:
>
> | Model       | Avg. Total Tokens | Total Latency | Extra Image Tokens | Extra Latency |
> | ----------- | :---------------: | :-----------: | :----------------: | :------------: |
> | DeepEyes-7B |        431        |      39s      |        190        | 51.5ms (~1.3%) |
>
> As the table shows, while iMCoT introduces extra visual tokens, the associated additional latency is negligible (~1.3%) compared to the total inference time. This overhead stems from the fast, parallel vision encoding pass, not the slower, autoregressive text generation. Thus, invoking the vision encoder with multiple times does not create a bottleneck.
>
> Furthermore, in real-world deployments, this overhead is minimized by modern serving infrastructure. Techniques such as KV Caching, Encode-Prefill-Decode (EPD) Disaggregation [1,2] can efficiently batch the processing of new image crops during the prefill phase.
>
> Furthermore, our training dynamics (Fig. 3, Stage 3) show the model ***learns to be efficient***. It learns to selectively invoke this scaling only when necessary, converging on a policy that balances accuracy and cost.
>
> ---
>
> > ## **W3: Detect first, then reason.**
>
> Thank you for proposing this insightful alternative. We believe that the improvements observed in DeeyEyes primarily stem from its iMCoT reasoning, where perception dynamically serves reasoning. As illustrated in the right panel of Fig.1, there is ***no obvious target object for the model to detect in the prompt***. Nevertheless, DeeyEyes autonomously constructs a reasoning path by sequentially searching and comparing subfigures to interpret and answer a vague question, demonstrating a level of abstraction and task generalization beyond object detection.
>
> To empirically substantiate this, we compared DeepEyes with a ***"detect-then-answer"*** baseline on the V* benchmark. Specifically, ***+self-crop*** bbox refers to a two-stage pipeline where the model first predicts the target object and then answers based on the crop; ***+gt*** bbox provides the ground-truth object crop, representing the upper bound of the reviewer's proposal (assuming perfect detection of the relevant object).
>
> | Model              |  V* acc. (%)  |
> | ------------------ | :------------: |
> | Qwen2.5-VL-7B      |      71.2      |
> | +self-crop bbox    |      75.8      |
> | +gt bbox           |      81.7      |
> | **DeepEyes** | **90.1** |
>
> The results confirm that simply concatenating high-quality local visual tokens—***even assuming perfect detection***—is insufficient. The performance gap highlights that DeepEyes succeeds not merely by "seeing clearly," but by reasoning iteratively, a capability that static visual concatenation cannot replace.
>
> ---
>
> ### **Reference**
>
> [1] ModServe: Modality- and Stage-Aware Resource Disaggregation for Scalable Multimodal Model Serving. arXiv:2502.00937.
>
> [2] HydraInfer: Hybrid Disaggregated Scheduling for Multimodal Large Language Model Serving. arXiv:2505.12658.

---

> ### Author Response · Authors · 2025-11-22
> **Response to Reviewer 3FYG - 2 / 2**
>
> > ## **W4: Low-resolution scenarios.**
>
> Thank you for the insightful question. To address this, we have added a new experiment evaluating fixed low-resolution inputs on the V* benchmark. We disable dynamic resolution (default setting by Qwen2.5-VL) and resize each image such that its longest edge is constrained to a specified value (while preserving aspect ratio). As shown in the table below, DeepEyes consistently outperforms Qwen2.5-VL-7B across all fixed resolutions, including the smallest 512px setting (57.6 vs. 32.5). These results demonstrate that DeepEyes’ gains are ***not dependent on high-resolution inputs***, and the method remains robust even under low-resolution scenarios. Dynamic resolution further improves performance but is not essential for the observed advantages.
>
> | Resolution (longest edge) |      512      |      1024      |      1536      |      2048      | Dynamic (avg. 2246) |
> | ------------------------- | :------------: | :------------: | :------------: | :------------: | :------------------: |
> | Qwen2.5-VL-7B             |      32.5      |      51.3      |      63.4      |      69.1      |         71.2         |
> | **DeeyEyes-7B**     | **57.6** | **74.9** | **76.4** | **82.7** |    **90.1**    |
> | *+improvement*          |   *+25.1*   |   *+23.6*   |   *+13.0*   |   *+13.6*   |      *+17.9*      |
>
> ---
>
> > ## **W5: MLLM-as-judge.**
>
> We apologize for the confusion. The MLLM served via vllm, which you correctly identified, acts purely as our ***Reward Model (RM)***. Its sole function is to calculate the final accuracy reward ($R_{acc}$) by comparing DeepEyes' generated answer against the ground truth. This RM was a deliberate design choice. To increase the RL learning difficulty, we adopted a "hard-to-answer, easy-to-verify" paradigm. We intentionally used free-form response tasks rather than simple multiple-choice tasks (which could be verified via rules). This necessitated a model-based RM to judge the semantic correctness of the answer—a task that is "easy-to-verify" for the RM.
>
> This RM provides only the final scalar reward. It provides zero SFT guidance or policy-level hints (e.g., where to ground). Therefore, our central claim holds: the "thinking with images" capability is learned natively and autonomously by DeepEyes, guided only by the final outcome. In practice, this RM call was fast and not a training bottleneck. We will clarify this mechanism in the revision.
>
> ---
>
> > ## **Q1: Transferring to more difficult tasks.**
>
> Thank you for this insightful and forward-looking question. This is a natural and exciting extension of our work. The core principle of DeepEyes—using end-to-end RL to incentivize a model to natively and iteratively query its perceptual understanding within a reasoning loop—is fundamentally modality-agnostic. We strongly believe this pipeline can be adapted:
>
> **For Video Tasks**: The "active perception" tool could evolve from spatial cropping to temporal grounding. An RL policy could learn when to pause, rewind, or "zoom in" on specific video segments to gather critical evidence for long-range reasoning, aligning with recent explorations in tool-augmented RL for video understanding [3].
>
> **For 3D Scenarios**: The challenge escalates to 3D spatial imagination from limited 2D views. The model could learn to natively generate and refine latent 3D representations during its reasoning steps, even without 3D priors. An RL policy could optimize when to exploit this internal 3D imagination to solve spatial tasks that are impossible from 2D cues alone, as explored in recent work on geometric-grounded reasoning [4].
>
> In both cases, the RL framework would learn an optimal policy for when to gather more perceptual data (e.g., temporal segments, latent 3D geometry) versus when to rely on its internal reasoning. We are optimistic that our approach can serve as a foundation for these more complex "thinking with X" pipelines.
>
> ---
>
> ### **Reference**
>
> [3] Thinking with videos: Multimodal tool-augmented reinforcement learning for long video reasoning. arXiv:2508.04416.
>
> [4] Think with 3D: Geometric Imagination Grounded Spatial Reasoning from Limited Views. arXiv:2510.18632.

---

### Official Review · Reviewer_8hFE · 2025-11-01

**Soundness:** 4
**Presentation:** 3
**Contribution:** 2
**Rating:** 6
**Confidence:** 3

**Summary:**

This paper introduces DeepEyes, a vision–language model (VLM) endowed with active perception capabilities through the integration of an image zoom-in tool. DeepEyes exhibits substantial performance gains over existing open-source and workflow-based baselines across a range of high-resolution visual tasks and reasoning benchmarks. Furthermore, the paper provides an in-depth analysis of the model’s training dynamics and emergent reasoning behaviors, uncovering patterns reminiscent of human visual cognition.

**Strengths:**

1. The paper is technically solid. The results (Tables 1–9) demonstrate consistent and notable improvements over relevant baselines in high-resolution perception (e.g., HR-Bench-8K), general reasoning, visual grounding, hallucination mitigation, and multimodal mathematical reasoning. The empirical rigor is further supported by comprehensive ablation studies and scaling experiments.
2. The paper conducts an insightful analysis of the training dynamics and emergent reasoning behaviors of **DeepEyes**, offering readers a deeper understanding of how active perception shapes visual reasoning.

**Weaknesses:**

1. Within the broader context of Agentic Reinforcement Learning (RL), this work can be regarded primarily as a **multimodal agents** equipped with a zoom-in tool, contributing limited methodological novelty.
2. The current implementation falls short of fully realizing the concept of **“Thinking with Images.”** This paradigm should encompass not only zoom-in operations but also more diverse capabilities—such as image editing, spatial manipulation, or compositional visual reasoning.
3. **Limited tool generalization.** Section 4.4 briefly asserts that the method generalizes to new tools; however, only a trivial extension to rotation is quantitatively validated. No systematic exploration of tool compositionality or multi-tool chaining—both central to agentic multimodal systems—is presented.

**Questions:**

1.Can the authors clarify how the model parameterizes and samples coordinates for zoom-in actions? Additionally, is any mechanism employed to prevent the repeated selection of non-informative or overlapping regions during active perception?

---

> ### Author Response · Authors · 2025-11-22
> **Response to Reviewer 8hFE**
>
> > ## **W1-3: Multimodal agent & limited tools.**
>
> We acknowledge the reviewer's valid point that the broader concept of "Thinking with Images" ideally encompasses diverse capabilities, including image editing and spatial manipulation. However, our work specifically targets **Active Perception through Reasoning** —the cognitive process where a model dynamically updates its visual working memory to resolve ambiguity. Unlike standard agents that treat tools as disjoint functional offloads, DeepEyes internalizes the "crop" operation to actively acquire focused visual information during the reasoning process.
>
> This is different from standard agents that simply call external APIs. In our framework, the visual operation is learned as an ***internal*** reasoning step. This allows the model to autonomously perform practical behaviors like searching for small objects or double-checking uncertain regions before answering. As demonstrated by our inclusion of the rotate tool in the paper, the framework is flexible and can adapt to new tools without retraining. We consider this a solid foundation for active perception and plan to integrate a wider range of tools and their compositionality in future work.
>
> ---
>
> > ## **Q1: Details of zoom-in actions.**
>
> DeepEyes inherits the coordinate-generation format from the Qwen2.5-VL series. The model outputs object-grounding actions in the following textual form:
>
> > {"name": "image_zoom_in_tool", "arguments": {"bbox_2d": [x1, y1, x2, y2], "label": "wetsuit"}}
>
> where (x1, y1) and (x2, y2) denote the upper-left and bottom-right corners of the bounding box. As illustrated in Fig. 7–12, the model directly generates these coordinates in text; if the format is valid, the numbers are parsed to form the cropping region for subsequent perception.
>
> DeepEyes is trained fully end-to-end with RL, and therefore we intentionally avoid injecting hand-crafted rules into the reasoning process. Designing such rules is challenging because visual reasoning is inherently multi-step and context-dependent. For example, determining what constitutes an “informative” region cannot be captured by simple deterministic criteria across diverse queries.
>
> Importantly, Fig. 9 shows that the reasoning pattern DeepEyes acquires after training resembles human visual analysis—progressive focusing, contextual checking, and purposeful refinement—rather than mechanically repeating similar perceptions. This aligns with the broader training dynamics: as RL progresses, DeepEyes learns to perform increasingly accurate and task-relevant zoom-in actions (see Sec. 4.3), indicating that effective grounding strategies emerge naturally without manual intervention.

---

### Meta-Review · Area_Chair_v8gn · 2026-01-05

**Summary:**

The workl introduces a vision–language model, termed DeepEyes, trained end to end with reinforcement learning to enable image-grounded reasoning. The model interleaves textual chain-of-thought with active visual perception, autonomously deciding when and where to attend by generating bounding boxes over image regions. These cropped regions are then reintroduced as observation tokens to support further reasoning. Training relies on agentic reinforcement learning using GRPO, sparse outcome-based rewards, and an additional conditional bonus that encourages correct solutions obtained through active perception. A dedicated data curation pipeline then identifies samples that are most likely to benefit from grounding-enhanced reasoning.

The paper recieved 4 reviews and all the reviewers were more or less positive about the work. All the reviewers praised the technical contribution of the paper as well as solid experimental validation with several benchjmarks being tested and strong results being reported.

**Reviewer Concerns:**

There were not any major concerns that were mentioned by the reviewers and the rebuttal, in my opinion, did a good job of answering any outstanding further concerns.

**Reviewer Scores:**

Although I cannot comment on score increases since all were already positive, I see atleast 1 reviewer moving from 6 to a 8. Overall I concur that the paper is strong and will be of great interest to the overall community at ICLR. I recommend acceptance.

---

### Decision · Program_Chairs · 2026-01-26

Accept (Poster)